# Embedding Trust: Semantic Isotropy Predicts Nonfactuality in Long-Form Text Generation

## Abstract

To deploy large language models (LLMs) in high-stakes application domains that require substantively accurate responses to open-ended prompts, we need reliable, computationally inexpensive methods that assess the trustworthiness of long-form responses generated by LLMs. However, existing approaches often rely on claim-by-claim fact-checking, which is computationally expensive and brittle in long-form responses to open-ended prompts. In this work, we introduce semantic isotropy—the degree of uniformity across normalized text embeddings on the unit sphere—and use it to assess the trustworthiness of long-form responses generated by LLMs. To do so, we generate several long-form responses, embed them, and estimate the level of semantic isotropy of these responses as the angular dispersion of the embeddings on the unit sphere. We find that higher semantic isotropy—that is, greater embedding dispersion—reliably signals lower factual consistency across samples. Our approach requires no labeled data, no fine-tuning, and no hyperparameter selection, and can be used with open- or closed-weight embedding models. Across multiple domains, our method consistently outperforms existing approaches in predicting nonfactuality in long-form responses using only a handful of samples—offering a practical, low-cost approach for integrating trust assessment into real-world LLM workflows.

## 1 Introduction

Large language models (LLMs) increasingly serve as front-line knowledge workers (Mayer et al., 2025). Yet the application of LLMs to high-stakes settings that require long-form responses to open-ended prompts is hamstrung by the fact that reliably ascertaining the trustworthiness of long-form text generated by LLMs remains challenging.

Without scalable, computationally-inexpensive methods that are able to reliably indicate the level of trustworthiness of long-form responses generated by a language model, deployment to high-stakes application domains that require substantively accurate responses to open-ended prompts will remain risky. Manual approaches are often infeasible at scale, and the standard stop-gaps—prompt engineering, system messages, and ensemble voting—remain brittle and expensive (Shorinwa et al., 2024). What is needed is a lightweight, model-agnostic, and data-agnostic method that is able to flag potentially untrustworthy text generations without needing to resort to slow, claim-by-claim fact-checking.

Prior work has attempted to gauge factuality in long-form natural language generation (NLG) by aligning atomic claims with a knowledge graph or by training auxiliary classifiers on annotated corpora (Wang et al., 2023). However, these approaches fall short in three important ways. First, they require structured references or costly ground-truth labels that may not exist in specialized domains. Second, they struggle with open-ended, multi-sentence answers where relevant facts are implicit rather than explicit (Nikitin et al., 2024). Third, their computational footprint grows rapidly with the number of sentences or claims, making them ill-suited to real-time applications (Jiang et al., 2024; Liu et al., 2024; Farquhar et al., 2024; Zhang et al., 2024; Manakul et al., 2023).

We address these limitations by introducing *semantic isotropy*—the degree of uniformity across normalized text embeddings on the unit sphere—and use it as a proxy for nonfactuality in long-form text generation. Intuitively, if a prompt admits a single, factually grounded explanation, independently sampled responses from a non-adversarial model should cluster tightly in embedding space (Qiu

**Figure 1: Diagrammatic Overview of the *Semantic Isotropy* Scoring Pipeline**. This illustration presents the experimental process used to derive a *semantic isotropy* score. For a given input, we sample $N$ i.i.d responses using a **Generative LLM**. Each response is fed through an **Embedding Model** which produces a vector representation in $R^D$. The matrix of $N \times D$ response embedding vectors is denoted as $E$. $E$ is transformed into a distance matrix $K_E^{\cos}$ using the cosine kernel, which is used to compute the isotropy score as $\mathcal{I}(K_E^{\cos}) = \text{vNE}(K_E^{\cos})/\log N$.

and Miikkulainen, 2024; Wang and Holmes, 2024). Conversely, when an LLM hallucinates, subtle changes in semantics and content pull the embeddings apart, inflating angular dispersion. To measure the semantic isotropy of a set of embeddings—and gauge the trustworthiness of the corresponding long-form text generations—we compute a *semantic isotropy score* according to the following simple and computationally inexpensive steps: We first generate a handful of responses, we then embed them with any off-the-shelf text encoder, and finally, we compute a semantic isotropy score by estimating the von Neumann entropy of the cosine kernel under the set of embeddings. No labels, fine-tuning, or hyperparameter search are required.

To enable broad evaluation of semantic isotropy as a proxy for nonfactuality—particularly across varying text lengths and at scale—we develop *Segment-Score*, a new factuality scoring method. *Segment-Score* is designed to address key limitations of existing approaches (Min et al., 2023): It is more efficient in terms of token usage, scales effectively to longer responses, and offers clearer, more consistent criteria for labeling statements as true or false. Using *Segment-Score*, we assess the reliability of *semantic isotropy scoring* in predicting nonfactuality long-form text generation. Computing semantic isotropy scores is model-agnostic and can be done with off-the-shelf embedding models, making this approach inexpensive and easy to use in practice.

In our empirical evaluation, we find that *semantic isotropy scoring* achieves state-of-the-art performance across relevant benchmarks. It does so robustly across a wide range of models, response lengths, and evaluation settings, demonstrating its effectiveness and generalizability as a lightweight and scalable proxy for trustworthiness.

To summarize, our key contributions are:

1. We introduce *semantic isotropy* and describe a simple and computationally inexpensive method for *semantic isotropy scoring* as a means to assess nonfactuality in long-form natural language generation.

2. We develop the *Segment-Score* protocol to generate and score datasets of long-form LLM generations for open-ended prompts and create a dataset of $1,182$ unique entities along with $\approx 65,450$ scored responses by three different models, each containing roughly between 25 and 60 distinct claims.

3. We demonstrate the efficacy of using *semantic isotropy scores* as a proxy for factual inconsistency in long-form natural language generation across a range of generative models, embeddings, and relevant experimental settings.

## 2 RELATED WORK

### 2.1 UNCERTAINTY QUANTIFICATION IN LARGE LANGUAGE MODELS

Early work on uncertainty quantification in LLMs largely involved analyzing token-level logits and derived distributions. OpenAI (2023) demonstrated the differences in token-level calibration dynamics of pre-trained and RLHF post-trained LLMs with Kirk et al. (2024) inferring that the latter process led to model miscalibration.

---

**Algorithm 1** SEGMENT-SCORE($t$)

---

**Require:** Topic $t$, LLM response $Y(t)$, reference document $\mathcal{D}(t)$, Oracle LLM $\mathcal{O}$
1: $S(t) \leftarrow \mathsf{Segmenter}_{\mathcal{O}}\big(Y(t)\big) = \langle s_1, s_2, \ldots, s_m \rangle$,    where $m \leftarrow |S(t)|$ ▷ Partition $Y(t)$ into $m$ atomic segments
2: Initialize $\mathbf{v} \leftarrow []$
3: **for** $i = 1$ **to** $m$ **do**
4:     $C_i \leftarrow \langle s_1, s_2, \ldots, s_{i-1} \rangle$                        ▷ Context before segment $s_i$
5:     $v_i \leftarrow \mathsf{Score}_{\mathcal{O}}\big(C_i, s_i, \mathcal{D}(t)\big)$            ▷ Verify $s_i$ against $\mathcal{D}(t)$ in context
6:     Append $v_i$ to $\mathbf{v}$
7: **end for**
8: $\phi \leftarrow \frac{1}{m} \sum_{i=1}^{m} \mathbf{1}(v_i = \mathsf{True})$           ▷ Fraction of segments verified as True
9: **return** $\phi$

---

On the other hand, Monte Carlo sampling-based approaches (Zhang, 2020; Malinin and Gales, 2021; Lin et al., 2024; Lakshminarayanan et al., 2017) have shown promise in estimating uncertainty in LLMs. However, a key challenge highlighted by these approaches is the need to standardize LLM responses to effectively compare them. In multiple-choice style discrete finite class output settings, studies have used a combination of Natural Language Inference (NLI) (Williams et al., 2018), Oracle LLMs (Band et al., 2024), and Embedding Models (Qiu and Miikkulainen, 2024; Grewal et al., 2025) to standardize model responses for precise class attribution. However, in NLI methods specifically, models have historically had limited context windows and several of the comparison algorithms grow quadratically in the number of inputs (Farquhar et al., 2024; Zhang et al., 2024; Manakul et al., 2023; Wang et al., 2024c; Chen et al., 2025). As a result, while the techniques used are in principle capable of addressing the semantic variations introduced by short-form responses, several limitations remain that hinder widespread adoption in practical settings.

A direction that addresses some of the pain points of NLI methods is the introspection of the model's internal state to estimate uncertainty. Kadavath et al. (2022) use the activations in the model's last hidden layer to train a linear classifier probe that indicates whether the model knows the answer to the prompt. Kossen et al. (2024) adopt a similar approach by developing a cheap predictive model on top of the LLM's internal state that is capable of estimating semantic entropy (Farquhar et al., 2024) without repeated sampling; Nikitin et al. (2024) extend this approach to analyze a graph representation and its Laplacian. Ji et al. (2024) use the state information to specifically predict the likelihood of hallucination while Ao et al. (2024) use latent representations in a multi-modal context to develop an uncertainty quantification method for both text and images simultaneously. While the success of these approaches justify the value of leveraging latent representations for uncertainty quantification, these existing methods generally do not scale well with increases in response size and model complexity.

Recently, several studies have made meaningful advances in addressing uncertainty quantification, specifically for longer-format settings. Long Uncertainty Quantification (LUQ) (Zhang et al., 2024) and INSIDE (EigenScore) (Chen et al., 2024) are most relevant to our method and serve as key benchmarks in our analysis. While LUQ uses NLI methods to calculate a similarity matrix between samples, the LUQ-Atomic variant that we benchmark against averages normalized entailment at a sentence / statement level, providing a much more fine-grained and robust measure of similarity. EigenScore also samples a batch of responses and computes the eigen-distribution over the batch of latent representations in the generating model.

## 2.2 FACTUALITY AND LONG-FORM NLG

While several NLP datasets are designed for multiple choice-style tasks (Joshi et al., 2017; Rajpurkar et al., 2016; Lin et al., 2022; Reddy et al., 2019), the options for open-ended long-form responses are much more limited. A key challenge is defining a suitable metric that is a meaningful measure of quality but also tractable to compute. Factuality, or the degree to which a response is supported by a ground truth, has been a go-to metric for several studies (Guo et al., 2022). Min et al. (2023) developed FactScore, a popular scoring algorithm for generating factuality scores using a ground truth text. FactScore operates by decomposing a text into a set of atomic facts by using an oracle

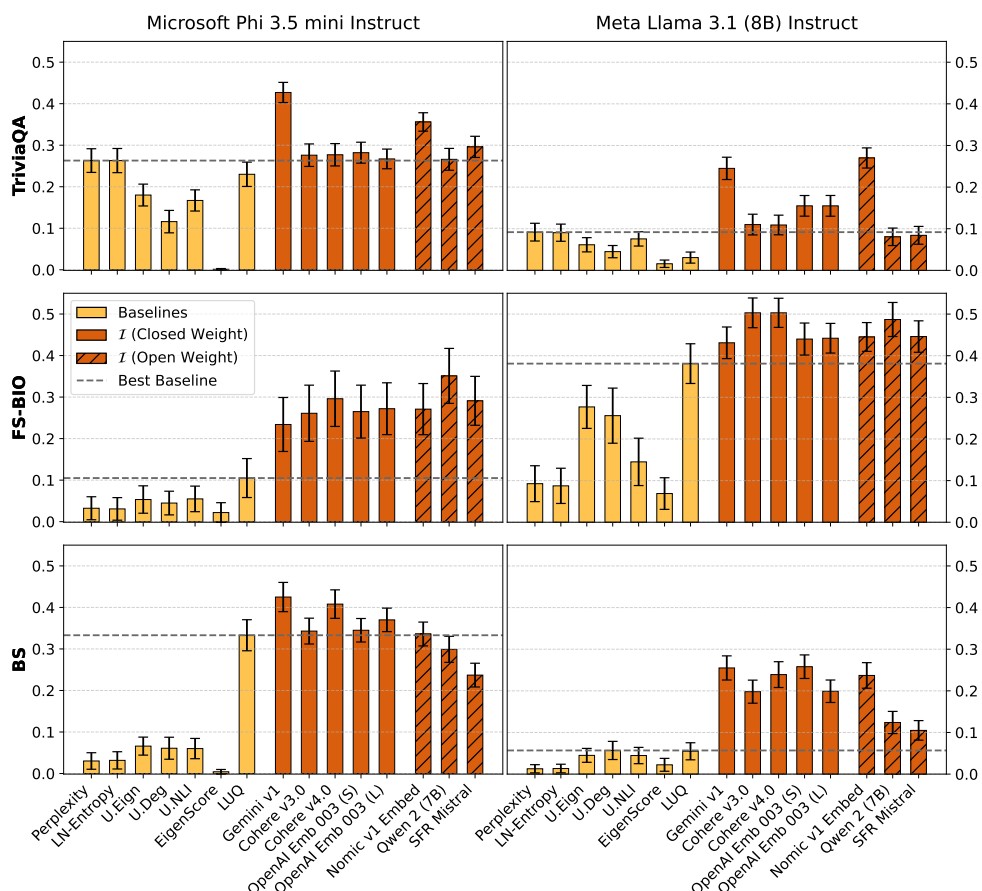

Figure 2: **Main Experimental Results**. Bar Charts comparing the performance (as measured by the $R^2$ of a linear model of Factuality $\sim$ Semantic Isotropy); implemented using various embedding models and benchmark uncertainty metrics on three datasets: TriviaQA Entities **[TriviaQA]** (Top Row), FactScore-Biographies **[FS-BIO]** (Middle Row) and CMU Book Summaries **[BS]** (Bottom Row); scored using the *Segment-Score* **(SS)** algorithm. Responses are $\approx$500 words. 1500 Boot-strapped samples are used to generate 1-SD error bars. Semantic isotropy $\mathcal{I}$ (ours) outperforms all other baselines for all embedding models.

model (LLM) and scoring each of these facts with the help of the ground truth reference document. This method has been applied to create datasets such as FactScore-Bio (Zhang et al., 2024), that operates on biographies taken from Wikipedia.

FactScore has given rise to several derivative methods such as LongFact & SAFE (Wei et al., 2024) (which uses dynamic web searches to find supporting references), FELM (Chen et al., 2023) (applied to Math and Reasoning problems), Multi-FAct (Shafayat et al., 2024) (multi-lingual setting), FactAl-ign (Huang and Chen, 2024) and LoGU (Yang et al., 2024b) (post-training / RLHF to incorporate uncertainty phrasing into model generations), and VeriScore (Song et al., 2024) (better precision by scoring verifiable facts only) to name a few. But most of these methods share key limitations of the FactScore framework, scaling poorly with response length and requiring several LLM calls per sample. We address some of these drawbacks through our method, *Segment-Score* (see Algorithm 1), that utilizes capabilities of more modern LLMs such as longer context windows and the ability to respond with structured outputs.

## 3 SEMANTIC ISOTROPY IN LONG-FORM LANGUAGE GENERATION

In this section, we introduce *semantic isotropy*, a measure of semantic variation in long-form text generations, and use it to motivate *semantic isotropy scoring*, a method that expresses the level of semantic isotropy in long-form responses and can serve as a proxy for nonfactuality.

## 3.1 Semantic Isotropy

We start with the conventional definition of isotropy. Intuitively, isotropy describes a state where a set of vectors or a distribution has no preferred direction—its properties are the same in all orientations. More formally, for a set of unit vectors, we can express this property as follows:

> **Definition (Isotropy).** *The column vectors $x_1, ..., x_N \in \mathbb{R}^D$ with $\|x_i\|_2 = 1$ for all $i \in \{1, ..., N\}$ are said to be isotropic if for the matrix $X = [x_1, ..., x_N]$: $X^\top X = I_N$.*

This condition means that, on average, the transformation represented by $X$ preserves directions and spreads uniformly across all directions in $\mathbb{R}^D$. When $X^\top X = I_N$, the linear transformation $x \to X^\top x$, for $x \in \mathbb{R}^D$, acts as an isometry on $\mathbb{R}^N$ embedded in $\mathbb{R}^D$ (with $N << D$ in our case). Geometrically, if the unit vectors $x_1, ..., x_N$ are isotropic, this implies that the unit vectors are uniformly dispersed across the unit sphere, giving us an intuitively meaningful measure of variation within $x_1, ..., x_N$.

We will now build on this intuition to define *semantic isotropy*. Let $\mathcal{R} = \{r_1, ..., r_N\} \in \mathcal{T}^N$ be a set of $N$ long-form NLG responses given a prompt $p$ and let $\Gamma_\theta(\cdot) : \mathcal{T} \to \mathbb{R}^D$ be an embedding model parameterized by neural network parameters $\theta$. We define $E = \{e_1, ..., e_N\}$ to be the collection of embedding vectors, where $e_i = \Gamma_\theta(r_i) \in \mathbb{R}^D$ is the embedding vector of response $r_i$, $\bar{e}_i \in \mathbb{R}^D$ is the corresponding normalized embedding vector, $\bar{e}_i = e_i / \|e_i\|$, and $\bar{E}$ is the matrix of normalized embedding vectors, $\bar{E} = [\bar{e}_1, ..., \bar{e}_N]^\top \in \mathbb{R}^{N \times D}$. The cosine kernel function for two embedding vectors, $e_i, e_j \in \mathbb{R}^D$, is then defined as

$$k^{\cos}(e_i, e_j) = [\bar{E}\bar{E}^\top]_{i,j} = \bar{e}_i^\top \bar{e}_j. \tag{1}$$

We denote the cosine kernel matrix for $E$ by $K_E^{\cos} \in \mathbb{R}^{N \times N}$.

With these preliminaries in place, we are now ready to formally define *semantic isotropy* to represent uniform dispersion of the normalized embedding vectors:

> **Definition (Semantic Isotropy).** *A set $\mathcal{R} = \{r_1, ..., r_N\}$ of long-form responses with corresponding embeddings $E = \{e_1, ..., e_N\}$ is semantically isotropic if $K_E^{\cos} = I_N$.*

## 3.2 Measuring Semantic Isotropy in Long-Form Language Generation

As defined, isotropy represents a strict condition that is either met or not met. Therefore, to use semantic isotropy in practice as a means of ascertaining nonfactuality in long-form natural language generation, we need to be able to gauge the *level* of isotropy of a given cosine kernel. We want to formalize the intuition that factuality corresponds to a high level of semantic alignment within a set of long-form generations (highly anisotropic), whereas uncertainty and non-facuality correspond to a high level of dispersion (very isotropic).

To estimate the level of isotropy of a set of long-form text generations, we use the von Neumann entropy (vNE) (von Neumann and Beyer, 2018) of the cosine kernel $K_E^{\cos}$, normalized to have trace $= 1$, which allows us to view the eigenvalues as defining a discrete probability distribution. Let $\bar{K}_E^{\cos} = K_E^{\cos}/\text{trace}(K_E^{\cos})$ and define

$$\text{vNE}(K_E^{\cos}) = -\text{trace}(\bar{K}_E^{\cos} \log \bar{K}_E^{\cos}). \tag{2}$$

To be precise, we view the above definition in the eigenspace of the positive semidefinite matrix $K_E^{\cos}$, that is, $\text{vNE}(K_E^{\cos}) = -\sum_i^N \lambda_i \log \lambda_i$, where $\lambda_1, ..., \lambda_N$ are the eigenvalues of $\bar{K}_E^{\cos}$. Note that $\text{vNE}(K_E^{\cos})$ attains its maximum value of $\log N$ for semantically (perfectly) isotropic responses (i.e., isotropic normalized embedding vectors) and is minimized at 0 when all embedding vectors are aligned (parallel). It thus measures the degree of dispersion of $\{\bar{e}_i\}_{i=1}^N$ on the unit sphere and serves as a measure of the level of isotropy within the set of text embeddings.

We are now ready to define the *semantic isotropy score* $\mathcal{I}(\cdot)$ that will serve as our proxy for factuality. Consider $N$ sampled long-form text generations in response to a prompt and define

$$\mathcal{I}(K_E^{\cos}) = \text{vNE}(K_E^{\cos})/\log N, \tag{3}$$

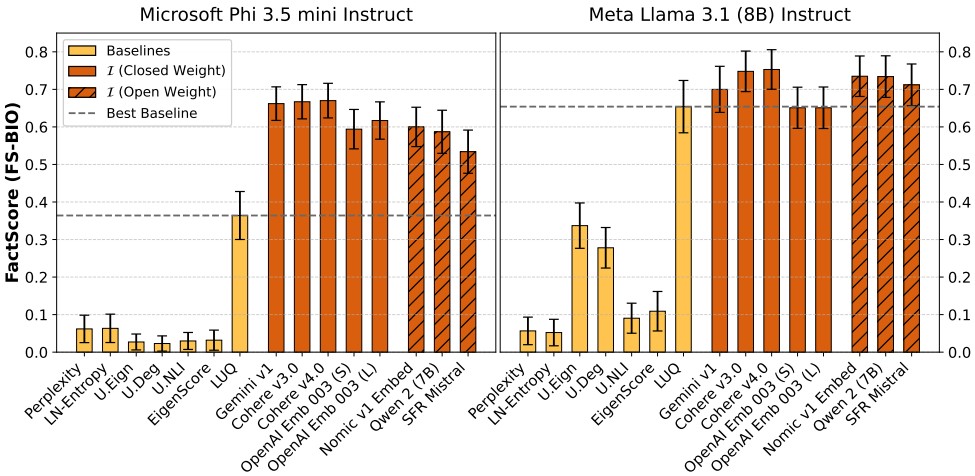

**Figure 3: FactScore Experimental Results**. Bar Charts comparing the performance (as measured by the $R^2$ of a linear model of factuality with semantic isotropy as the explanatory variable; implemented using various embedding models and benchmark uncertainty metrics on the **FS-BIO** dataset using the *FactScore* (**FS**) scoring algorithm. Left hand side: Benchmark UQ metrics. Right Hand Side: semantic isotropy ($\mathcal{I}$) implemented using various embedding models. Same experimental setting as in Figure 2. We observe that our method's performance is robust to the scoring scheme.

with $0 \leq \mathcal{I}(K_E^{\cos}) \leq 1$. Under this definition, (i) a lower degree of semantic isotropy (a semantic isotropy score closer to 0) is associated with stronger alignment/lower dispersion and consequently higher certainty and factuality, whereas (ii) a higher degree of semantic isotropy (a higher semantic isotropy score closer to 1) is associated with weaker alignment/higher dispersion and consequently lower certainty and factuality. In a nutshell: *The lower* the semantic isotropy score, *the more trustworthy* the generating model's long-form responses to the prompt are.

We also considered alternative matrix measures—like the Frobenius norm, $\|K_E^{\cos}\|_F$, and the log determinant, $\log(\det(K_E^{\cos}))$—for estimating the level of embedding isotropy and found that using the von Neumann entropy works best (see Table 1). This is consistent with the findings of Nikitin et al. (2024). When assessing the efficacy of other measures of isotropy, we found that several other measures performed well, suggesting that even under different measurement approaches, isotropy is a robust proxy for nonfactuality.

## 4 EMPIRICAL EVALUATION

In this section, we conduct an empirical evaluation to assess whether semantic isotropy scores can serve as effective a proxy for nonfactuality in long-form natural language generation with LLMs. First, we will describe the creation of a ground truth evaluation dataset using *Segment-Score* and present our evaluation metrics and benchmarks. We will then present our main findings about the efficacy of semantic isotropy scoring as a means to predict nonfactuality in long-form text generation.

### 4.1 CREATION OF A DATASET OF LONG-FORM RESPONSES TO OPEN-ENDED PROMPTS

To construct a dataset of long-form response to open-ended prompts, we must select a corpus of entities such that each is associated with an underlying ground truth reference document. These entities are used to create a set of open-ended prompts that can be fed into an LLM to generate long-form responses. We use three sources—*FactScore-Bio* [**FS-BIO**] (182 unique entities) (Min et al., 2023; Zhang et al., 2024), **TriviaQA** (Joshi et al., 2017) (1,000 unique entities) and CMU Book Summaries [**BS**] (509 unique entities) (Bamman and Smith, 2013). All three datasets use Wikipedia as their ground truth to source reference documents. For **TriviaQA**, we exclude all entities that correspond to days, dates or numerical values and only select those that match the title of the underlying Wikipedia page. Of the 5,245 qualifying TriviaQA entities across the training and validation set, we randomly select a subset of 1,000 entities to create open-ended prompts for long-form text generation. For *Book Summaries* (**BS**), we constrain the dataset to entities where the

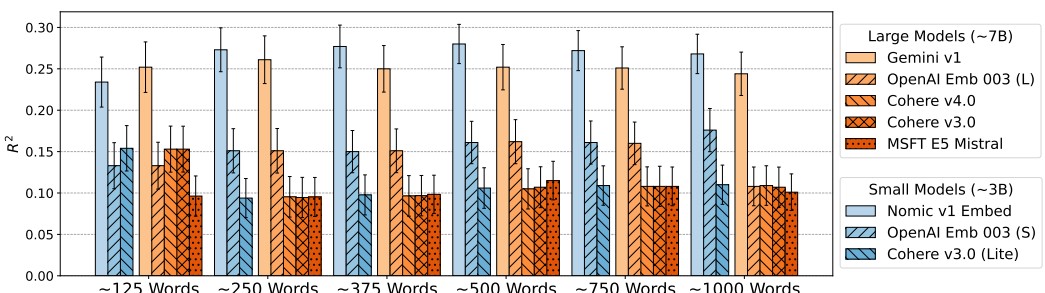

**Figure 4: Performance by Response Length**: Comparing *Semantic Isotropy* in terms of explained variance ($R^2$) of Factuality (**SS**) across various response lengths, contrasted by model size: Small (up to $3B$ parameters) and Large ($\approx 7B$ or more parameters). 1500 Bootstrapped samples are used to generate 1-SD error bars.

underlying reference text is at least 5,000 characters and the publish date is known, and is on or after Jan 1, 2000. Of a total of 16,559 entries, we select 509 entities that meet this criteria.

To generate long-form responses, we use Meta Llama 3.1 8B Instruct (Touvron et al., 2023), Microsoft Phi 3.5 Mini Instruct (Abdin et al., 2024), and OpenAI GPT 4.1 Mini (OpenAI, 2023) to generate responses for **TriviaQA** and **FS-BIO**. For **BS**, we only generate responses using Llama 3.1 and Phi 3.5 Mini. For **TriviaQA** we sample up to $k = 20$ responses while for **FS-BIO** and **BS** we sample $k = 10$ responses, targeting approximately 500 words for each response. For **TriviaQA** using Llama 3.1 8B Instruct, we also generate a longer dataset of approximately $1,000$ words and derived datasets of intermediate word counts of approximately 125, 250, 375, 500, and 750 words by truncating the $1,000$ word variant to the nearest sentence. Model inference is run with temperature $\tau = 0.7$ and FP16 quantization using the vLLM framework (Kwon et al., 2023). Finally, we compute factuality scores for the datasets using the *Segment-Score* (**SS**) procedure (Algorithm 1). Implementation details and prompts are presented in Appendix B.3. The dataset and Segment-Score algorithm are released to allow future benchmarking by the research community.

We also generate *FactScore* (**FS**; Min et al., 2023) scores for the **FS-BIO** datasets from each model. This allow us to benchmark our scoring algorithm (**SS**) to the existing state-of-the-art method, and we can show that our results hold robustly. See Appendix C.2 for a comparison of the scoring methods. We use GPT 4.1 Mini (OpenAI, 2023) as the oracle LLM in both algorithms.

## 4.2 EVALUATION METRICS

To assess the performance of our methods and compare them to appropriate baselines from the literature, we measure the degree of the relationship between each individual measure and the average factuality of a given topic, aggregated across all responses. First, we selected a set of embedding models that rank highly on the MTEB Benchmark (Muennighoff et al., 2023) to use in computing our isotropy scores. We measure predictiveness of these scores using $R^2$, the explained variance of a linear model with factuality scores as the dependent variable and isotropy scores as the independent variable. To estimate error bounds, we perform bootstrap sampling and generate 1-$\sigma$ error bars for each measure. We compare *Semantic Isotropy* to the following baseline measures: **Perplexity** - Exponential of the average negative log-likelihood of the generation (Ren et al., 2023); **LN-Entropy** - Length Normalized Entropy (Malinin and Gales, 2021); **U.Eign** - Sum of Eigenvalues of the Graph Laplacian, **U.Deg** - Average pairwise distance measured using an NLI model (Lin et al., 2024); **U.NLI** - NLI uncertainty derived using SelfCheckNLI (Manakul et al., 2023); **Semantic Entropy** - Entropy of the probability distribution of semantic clusters over the responses (Farquhar et al., 2024); **LUQ-Atomic** - Long Uncertainty Quantification using atomic facts via LLM based fact extraction (Zhang et al., 2024); **EigenScore** - the log-determinant of the centered embedding matrix of the generating model's internal state activations over a set of sampled responses Chen et al. (2024).

While relevant to our study, we do not include comparisons with **Kernel Language Entropy (KLE)** (Nikitin et al., 2024) and **Graph Longform Uncertainty** (Jiang et al., 2024). We found that the KLE computation was highly numerically unstable given the degree of semantic entailment overlap among the sampled long-form responses and did not produce meaningful results. For Graph Uncertainty, the

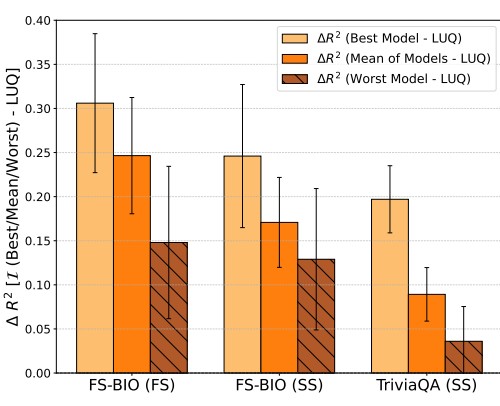

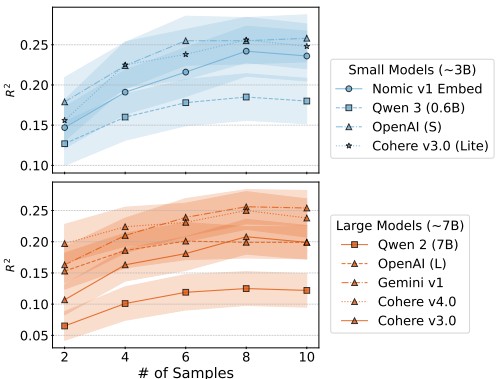

(a) **Model Agnostic Performance Comparison**: Performance difference between current state-of-the-art **LUQ-Atomic** (Zhang et al., 2024) and semantic isotropy based on different embedding models across each dataset using Phi 3.5 Mini Instruct as the generator model. In each case, all versions of semantic isotropy are more predictive than LUQ. 1500 bootstrapped samples are used to generate 1-SD error bars.

(b) **Performance by # of Samples**: Line Charts comparing the performance of semantic isotropy by number of sampled responses on **FS-BIO (SS)** dataset of $\approx 500$ words, contrasted by model size: Small (up to $3B$ parameters) and Large ($\approx 7B$ or more parameters). 1,500 Bootstrapped samples are used to generate 1-SD error regions.

Figure 5: **Ablation Studies:** Comparing the performance of semantic isotropy when varying (a) Embedding Model Used [Left] and (b) Number of samples used to measure Isotropy [Right].

distinct claim union algorithm proved computationally difficult to implement at the response lengths considered in our study. For context, a typical set of responses had over 600 distinct claims, which implied 6,000 LLM calls to measure claim to response entailment for one topic, assuming $N = 10$.

**Sampling the appropriate embedding.**    Unlike closed-weight models that return just an embedding vector, for open-weight models we select the activations corresponding to the last token in the final hidden state, except for Nomic v1 (Nussbaum et al., 2025) where we take the average of the activations over the token dimension instead. For further details, see Section 5.

### 4.3 MAIN RESULT: FACTUALITY IN LONG-FORM RESPONSES TO OPEN-ENDED PROMPTS

We present our main empirical results in Figure 2 (see also Table 4 in Appendix A). We find that semantic isotropy scores are superior to existing methods in predicting nonfactuality as per the **(SS)** algorithm on the **FS-BIO**, **TriviaQA** and **BS** datasets. Using $R^2$ (i.e., the OLS-explained variance) as our performance measure, $\mathcal{I}$ outperforms all existing metrics, in most cases by a wide margin. We see some variance in the performance under different embedding models, and discuss this finding in more detail in Section 5. While certain embedding models are better suited to specific tasks, semantic isotropy scoring outperforms existing state-of-the-art methods across the board. Notably, the `Nomic V1` embedding (Nussbaum et al., 2025) demonstrates exceptional performance on all datasets, occasionally surpassing even larger and closed-source models such as Gemini (Team, 2024) and OpenAI Embeddings (OpenAI, 2023).

### 4.4 FACTSCORE EVALUATION

To ensure that our results are not merely an artifact of the *Segment-Score* method, we also evaluate semantic isotropy scoring under the *FactScore* (**FS**) algorithm applied to the **FS-BIO** dataset (see Figure 3). We find that the results mirror those in Figure 2 even improving the predictive performance of semantic isotropy scoring. This consistency across scoring schemes highlights the robustness and generalizability of semantic isotropy as a measure of nonfactuality. A more detailed comparison of the two scoring methods can be found in Appendix C.2.

### 5 SENSITIVITY STUDIES

**Effect of Response Length.**    One important study we undertake is to understand the relationship between measure performance and response length. While our main study focuses on $\approx 500$ word

responses, we explore the relationship between semantic isotropy and factuality for shorter and longer texts as well. We outline these results in Figure 4. While certain models perform better (Gemini and Nomic v1), performance is generally consistent across the board. OpenAI's Text Embedding 3 is a notable exception, with performance consistently increasing with response length, independent of model size.

**Effect of Choice of Embedding Model.** We analyze how the size and type of embedding model impact performance. As shown in Figure 5a, *Semantic Isotropy* consistently outperforms LUQ across all three datasets/scoring methods and embedding models. Notably, even the least effective embedding model yields improvements over LUQ, while the best performs substantially better. These results highlight the robustness of *Semantic Isotropy*, demonstrating its effectiveness regardless of the embedding model used. We compare small and large variants of closed source models such as those provided by OpenAI and Cohere. In both cases, the performance difference is negligible and likely attributable to sampling noise. This finding demonstrates that semantic isotropy is robust even at small model sizes.

**Effect of Number of Sampled Responses.** Figure 5b illustrates how performance scales with the number of sampled responses. Encouragingly, semantic isotropy scoring does not require many samples ($\approx$ 6-8) to achieve comparable performance to its implied asymptotic level (at $N = 10$ samples), across both small and large models. While the method is generally scalable, this result confirms that it can deliver consistent truthfulness indicators with minimal overhead.

**Effect of Generator Capability Level.** In addition to the results discussed in Section 4.3, where we used long-form responses generated by Phi 3.5 Mini Instruct and Llama 3.1 8B Instruct, we also considered long-form responses generated by a more capable, closed-weight model, GPT 4.1 Mini. Applying the *Segment-Score* method to long-form responses for FS-BIO and TriviaQA, we find that the absolute performance of semantic isotropy scoring as well as its relative performance compared to the baselines significantly increases on FS-BIO but decreases on TriviaQA. On TriviaQA, semantic isotropy scoring achieves a higher $R^2$ than all baselines using `Nomic v1` embeddings, but the best absolute $R^2$ value is barely above 10%, implying that almost none of the variation in the factuality scores can be explained by any of the methods used. Since, for FS-BIO, semantic isotropy scoring performs better on long-form responses generated by GPT 4.1 Mini than on long-form responses generated by Phi 3.5 Mini Instruct and Llama 3.1 8B Instruct, the poor performance (across methods) on TriviaQA is unlikely due to increased model capability per se and more likely due to dataset idiosyncrasies, such as inherent entity ambiguities, or model pre-training artifacts. We present a detailed analysis and discussion in Appendix C.1.

**Computational Considerations.** One of the key drawbacks of existing methods such as LUQ and Graph Uncertainty is the $\mathbf{O}(MN^2)$ complexity, where $M$ is the average number of facts or segments per response and $N$ is the number of sampled responses. Even for Semantic Entropy and derived methods, the computation of entailment scores is $\mathbf{O}(N^2)$. Entailment scores are also computationally expensive as premises and hypothesis cannot be naively entailed in a vectorized fashion. In contrast, while semantic isotropy is naively $\mathbf{O}(N^2)$, where $N$ is the number of

Table 1: **Comparison of different isotropy measures**: Analysis of various isotropy measures, each of which can represent the isotropy condition. Comparing explained variance ($R^2$) of Factuality on $\approx$ 500 word length responses generated using Phi 3.5 Mini Instruct on the **TriviaQA (SS)** dataset, across various embedding models. 1,500 Bootstrapped samples are used to generate 1-SD error bars.

| Model | Frobenius | Inv. Trace | LogDet | vNE(OURS) |
|---|---|---|---|---|
| Gemini | 0.412 ± 0.02 | 0.376 ± 0.02 | **0.431 ± 0.02** | 0.43 ± 0.02 |
| Nomic V1 | 0.326 ± 0.02 | 0.329 ± 0.02 | **0.393 ± 0.02** | 0.354 ± 0.02 |
| OpenAI (L) | 0.257 ± 0.02 | 0.208 ± 0.02 | 0.255 ± 0.02 | **0.266 ± 0.02** |
| OpenAI (S) | 0.274 ± 0.02 | 0.167 ± 0.02 | 0.241 ± 0.02 | **0.279 ± 0.02** |
| Cohere v4.0 | 0.272 ± 0.02 | 0.222 ± 0.02 | 0.259 ± 0.02 | **0.276 ± 0.02** |
| Cohere v3.0 | 0.272 ± 0.02 | 0.221 ± 0.02 | 0.261 ± 0.02 | **0.276 ± 0.02** |
| Qwen 2 (7B) | **0.27 ± 0.02** | 0.169 ± 0.02 | 0.232 ± 0.02 | 0.268 ± 0.02 |
| SFR Mistral (7B) | 0.291 ± 0.02 | 0.186 ± 0.02 | 0.268 ± 0.02 | **0.297 ± 0.02** |

samples, the ability to compute it in a parallelized vectorized way yields significant performance improvements. We generally find that for scoring one batch of $N = 20$ responses for a given topic of $\approx$ 500 words using $\mathcal{I}$ requires $1.8 \pm 0.05$ seconds on a V100 GPU (amortizing the overhead over 20 trials), compared to $302 \pm 48$ seconds for LUQ-Atomic, using models of comparable parameter counts—that is, `Nomic v1` for $\mathcal{I}$ and `Deberta V3 Large (MNLI)` (Manakul et al., 2023) for LUQ.

**Effect of Choice of Isotropy Measure.**    In Table 1 we ablate a variety of alternative measures of isotropy, among them the Frobenius norm $\|K_E^{\cos}\|$, $\log(\det(K_E^{\cos}))$ and $\text{trace}(K_E^{\cos})^{-1}$. We find that von Neumann entropy-based isotropy scoring performs best overall, outperforming other measures on average across different embedding model choices and dataset settings. However, most of these measures demonstrate nearly equivalent performance, further underscoring the robustness of semantic isotropy as a proxy for nonfactuality.

**Coherence of Segment-Score.**    While *Segment-Score* offers several practical benefits, it remains critical to establish that it is ultimately a sound proxy for factuality. To calibrate FactScore, Min et al. used a cohort of human evaluators to match against. Given the size and span of our datasets, we opted for an LLM-assisted workflow using several SOTA oracle models such as GPT 4.1 (OpenAI, 2023) or Claude 4 (Anthropic, 2024). For each response, we sample up to 5 True/False labels for each segment within,

Table 2: **Segment-Score Coherence Test**: Comparison of label agreement of *Segment-Score* classifications and those obtained using a majority voting scheme. For each dataset, we randomly sample 30 entities and score their response corpus using GPT 4.1, Claude 4 Sonnet and DeepSeek, taking 5 samples each. Scores are in percentage (%).

| Dataset | Gen. Model | OpenAI (GPT 4.1) | Claude (4 Sonnet) | DeepSeek |
|---------|-----------|--------|--------|----------|
| **BS** | Llama 3.1 | **89.24** | 81.33 | 75.87 |
|  | Phi 3.5 Mini | **91.28** | 86.13 | 80.38 |
| **FS-BIO** | Llama 3.1 | 83.92 | 86.13 | **86.39** |
|  | Phi 3.5 Mini | 76.98 | 86.67 | **88.30** |

given the reference document. We use the majority of these labels as our proxy for a true class label. In Table 2, we provide the matching accuracy against the one-shot labels generated by *Segment-Score*. The high degree of agreement indicates that the *Segment-Score* algorithm can be effectively utilized to proxy factuality. In Appendix C.2, we elaborate on the study relating the scores obtained by *Segment-Score* and *FactScore* algorithms on the **FS-BIO** dataset.

**Effect of Choice of Embedding Aggregation.**    Our experimental results require an embedding vector $z_i \in \mathbb{R}^D$. Closed-weight models return a single vector per response and allow the user to configure embedding side ($D$) as an input. We use the model default for Gemini (768), OpenAI (1536) and Cohere (1024). However, Open-weight embedding models like Qwen2 (Yang et al., 2024a) and Mistral (Meng et al., 2024) specify using the last token embedding in the final hidden state. This can be found in their reference implementations on HuggingFace. In contrast, for Nomic V1 (Nussbaum et al., 2025), authors do not recommend a pooling method and encourage users to find methods that work well for their use case. We find that the last token embedding method yields a weak semantic isotropy score. Instead, we use the mean pooling over the token dimension. Given model activations of the last hidden layer as a matrix $E \in \mathbb{R}^{L \times D}$, where $L$ is the number of tokens in the response and $D$ the embedding dimension, we produce $Z_{\texttt{nomic}} = \texttt{mean}(E, \texttt{dim=1})$ such that $Z_{\texttt{nomic}} \in \mathbb{R}^D$. In Figure 6, we show the difference between a highly trustworthy and a low trustworthy topic by mean, max and last token pooling for Nomic v1.

## 6   CONCLUSION

In this work, we introduced *Semantic Isotropy*, proposed a computationally inexpensive method for semantic isotropy scoring, presented *Segment-Score* for cheap long-form ground truth data generation, and demonstrated that semantic isotropy scores yield state-of-the-art factuality prediction in long-form natural language generation. We found that semantic isotropy is highly predictive of nonfactuality in long-form LLM responses to open-ended prompts and that semantic-isotropy scores outperform alternative approaches—in most cases by a large margin. We also found that, in most of the cases considered in our analysis, semantic isotropy scoring is robust to the choice of embedding model, the choice of isotropy measure, the response length, and the number of samples used. We release code to reproduce the empirical evaluation and implement the *Segment-Score* method, and we make the *Segment-Score*-annotated dataset publicly available to facilitate further research into assessing factuality in long-form text generation. Our findings suggest that semantic isotropy is a simple and yet effective predictor of nonfactuality and open the door to more reliable and cost-effective evaluation of long-form LLM text generation at scale.

## REPRODUCIBILITY STATEMENT

We have made a significant effort to ensure the reproducibility of our results. An anonymized implementation of our method is provided at anonymous.4open.science/r/semantic_isotropy-C927, which includes training, evaluation, and analysis scripts. The experimental setup—including hyperparameters, model configurations, and sampling parameters are described in Section 4.1. All datasets used in our experiments are publicly available, and we additionally provide scripts for data preparation. Our experiments rely on open-weight language models and certain API-only models; instructions for reproducing results with both included in the README.md file in the released code. Infrastructure and compute requirements are documented in Appendix B.1. Finally, we intend to release the dataset constructed for this study upon completion of the review process, enabling direct benchmarking of our results.

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

# APPENDIX

## TABLE OF CONTENTS

## A  FULL EXPERIMENTAL RESULTS

| Measure / Metric | FactScore-BIO (FactScore) | | |
| --- | --- | --- | --- |
| | **Llama 3.1 8B** | **Phi 3.5 Mini** | **GPT 4.1 Mini** |
| Perplexity (Ren et al., 2023) | $0.0578 \pm 0.04$ | $0.0622 \pm 0.04$ | $0.0158 \pm 0.02$ |
| LN Entropy (Malinin and Gales, 2021) | $0.0535 \pm 0.04$ | $0.0608 \pm 0.04$ | $0.00625 \pm 0.009$ |
| U.Eign (Lin et al., 2024) | $0.344 \pm 0.06$ | $0.0279 \pm 0.02$ | $0.0185 \pm 0.02$ |
| U.Deg (Lin et al., 2024) | $0.28 \pm 0.05$ | $0.0224 \pm 0.02$ | $0.0255 \pm 0.02$ |
| U.NLI (Manakul et al., 2023) | $0.0916 \pm 0.04$ | $0.0299 \pm 0.02$ | $0.0263 \pm 0.02$ |
| EigenScore (Chen et al., 2024) | $0.107 \pm 0.05$ | $0.0309 \pm 0.03$ | N/A |
| LUQ (Zhang et al., 2024) | $0.657 \pm 0.07$ | $0.367 \pm 0.06$ | $0.192 \pm 0.06$ |
| Semantic Entropy (Farquhar et al., 2024) | $0.136 \pm 0.06$ | $0.00868 \pm 0.01$ | $0.0152 \pm 0.02$ |
| $\mathcal{I}$: Gemini[†] (Team, 2024) | $0.7 \pm 0.06$ | $0.661 \pm 0.05$ | $0.48 \pm 0.08$ |
| $\mathcal{I}$: OpenAI Small[†] (OpenAI, 2023) | $0.65 \pm 0.06$ | $0.593 \pm 0.05$ | $0.38 \pm 0.08$ |
| $\mathcal{I}$: OpenAI Large[†] (OpenAI, 2023) | $0.649 \pm 0.06$ | $0.614 \pm 0.05$ | $0.376 \pm 0.08$ |
| $\mathcal{I}$: Nomic v1 (Nussbaum et al., 2025) | $0.737 \pm 0.05$ | $0.6 \pm 0.05$ | $0.547 \pm 0.08$ |
| $\mathcal{I}$: Qwen 2 (7B) (Yang et al., 2024a) | $0.733 \pm 0.05$ | $0.589 \pm 0.05$ | $0.569 \pm 0.08$ |
| $\mathcal{I}$: SF Mistral (Meng et al., 2024) | $0.714 \pm 0.05$ | $0.533 \pm 0.06$ | $\mathbf{0.607 \pm 0.08}$ |
| $\mathcal{I}$: Mistral (E5) (Wang et al., 2024a) | $0.714 \pm 0.06$ | $0.508 \pm 0.06$ | $0.604 \pm 0.08$ |
| $\mathcal{I}$: Cohere v4.0[†] (AI, 2025) | $0.747 \pm 0.06$ | $\mathbf{0.671 \pm 0.05}$ | $0.5 \pm 0.08$ |
| $\mathcal{I}$: Cohere v3.0[†] (AI, 2023) | $\mathbf{0.75 \pm 0.06}$ | $0.665 \pm 0.05$ | $0.503 \pm 0.08$ |
| $\mathcal{I}$: Cohere v3.0 (Lite)[†] (AI, 2023) | $0.749 \pm 0.05$ | $0.661 \pm 0.05$ | $0.503 \pm 0.08$ |
| $\mathcal{I}$: Qwen 3 (0.6B) (Yang et al., 2025) | $0.719 \pm 0.06$ | $0.525 \pm 0.06$ | $0.593 \pm 0.08$ |
| $\mathcal{I}$: Qwen 3 (4B) (Yang et al., 2025) | $0.713 \pm 0.06$ | $0.588 \pm 0.05$ | $0.554 \pm 0.08$ |
| $\mathcal{I}$: Qwen 3 (8B) (Yang et al., 2025) | $0.709 \pm 0.06$ | $0.561 \pm 0.06$ | $0.528 \pm 0.07$ |

**Table 3: Main Experimental Results: FactScore**. Comparison of semantic isotropy with other uncertainty metrics on **FactScore-Bio** using FactScore. Each dataset column is subdivided by base model: Llama 3.1 8B, Phi 3.5 Mini, and OpenAI 4.1 Mini. Values are $R^2$ (explained variance) of a simple linear model of Factuality $\sim$ Isotropy score. 1500 Bootstrapped samples are used to generate 1-SD error bars. †denotes API-only models. Note that Eigenscore values are not generated for GPT 4.1 Mini as it is a black-box model and the internal layer activations are not available.

## B  EXPERIMENTAL DETAILS

### B.1  INFRASTRUCTURE REQUIREMENTS

Our work was performed using Google Cloud Platform compute to run inference on the white-box models using a single node with up to 4 NVIDIA Tesla V100 GPUs. GPT 4.1 Mini was used as the Oracle LLM in Algorithm 1. OpenAI `text-embedding-3-small`, `text-embedding-3-large`, Gemini `gemini-embedding-001`, Cohere

| Measure / Metric | TriviaQA Entities | | | FactScore-BIO | | | Book Summaries | |
|---|---|---|---|---|---|---|---|---|
| | Llama 3.1 8B | Phi 3.5 Mini | GPT 4.1 Mini | Llama 3.1 8B | Phi 3.5 Mini | GPT 4.1 Mini | Llama 3.1 8B | Phi 3.5 Mini |
| Perplexity (Ren et al., 2023) | $0.092 \pm 0.02$ | $0.263 \pm 0.03$ | $0.00074 \pm 0.003$ | $0.0933 \pm 0.04$ | $0.0305 \pm 0.03$ | $0.0137 \pm 0.007$ | $0.0128 \pm 0.01$ | $0.0302 \pm 0.02$ |
| LN Entropy (Malinin and Gales, 2021) | $0.091 \pm 0.02$ | $0.26 \pm 0.03$ | $0.0306 \pm 0.01$ | $0.0872 \pm 0.04$ | $0.0316 \pm 0.03$ | $0.0241 \pm 0.02$ | $0.0134 \pm 0.01$ | $0.0317 \pm 0.02$ |
| U.Eign (Lin et al., 2024) | $0.062 \pm 0.02$ | $0.181 \pm 0.03$ | $0.00166 \pm 0.002$ | $0.275 \pm 0.05$ | $0.0529 \pm 0.03$ | $0.0155 \pm 0.02$ | $0.045 \pm 0.02$ | $0.066 \pm 0.02$ |
| U.Deg (Lin et al., 2024) | $0.0457 \pm 0.01$ | $0.117 \pm 0.03$ | $0.00286 \pm 0.003$ | $0.255 \pm 0.07$ | $0.0452 \pm 0.03$ | $0.0254 \pm 0.02$ | $0.0568 \pm 0.02$ | $0.0609 \pm 0.03$ |
| U.NLI (Manakul et al., 2023) | $0.0767 \pm 0.02$ | $0.167 \pm 0.03$ | $0.00164 \pm 0.002$ | $0.144 \pm 0.06$ | $0.0552 \pm 0.03$ | $0.0128 \pm 0.01$ | $0.0446 \pm 0.02$ | $0.0601 \pm 0.02$ |
| EigenScore (Chen et al., 2024) | $0.0158 \pm 0.009$ | $0.00131 \pm 0.002$ | N/A | $0.0683 \pm 0.04$ | $0.021 \pm 0.02$ | N/A | $0.0223 \pm 0.02$ | $0.00422 \pm 0.006$ |
| LUQ (Zhang et al., 2024) | $0.03 \pm 0.01$ | $0.231 \pm 0.03$ | $0.1 \pm 0.02$ | $0.381 \pm 0.05$ | $0.104 \pm 0.05$ | $0.116 \pm 0.04$ | $0.0548 \pm 0.02$ | $0.333 \pm 0.04$ |
| Semantic Entropy (Farquhar et al., 2024) | $0.0291 \pm 0.01$ | $0.0817 \pm 0.02$ | $0.00242 \pm 0.003$ | $0.156 \pm 0.07$ | $0.0298 \pm 0.02$ | $0.00692 \pm 0.009$ | $0.0187 \pm 0.01$ | $0.0342 \pm 0.02$ |
| $\mathcal{I}$: Gemini[†] (Team, 2024) | $\mathbf{0.244 \pm 0.03}$ | $\mathbf{0.427 \pm 0.02}$ | $0.028 \pm 0.01$ | $0.435 \pm 0.04$ | $0.235 \pm 0.07$ | $0.342 \pm 0.04$ | $0.255 \pm 0.03$ | $\mathbf{0.425 \pm 0.04}$ |
| $\mathcal{I}$: OpenAI Small[†] (OpenAI, 2023) | $0.156 \pm 0.02$ | $0.282 \pm 0.02$ | $0.00196 \pm 0.003$ | $0.441 \pm 0.04$ | $0.265 \pm 0.07$ | $0.314 \pm 0.04$ | $0.258 \pm 0.03$ | $0.345 \pm 0.03$ |
| $\mathcal{I}$: OpenAI Large[†] (OpenAI, 2023) | $0.155 \pm 0.02$ | $0.269 \pm 0.02$ | $0.00215 \pm 0.003$ | $0.44 \pm 0.03$ | $0.268 \pm 0.06$ | $0.315 \pm 0.04$ | $0.199 \pm 0.03$ | $0.37 \pm 0.03$ |
| $\mathcal{I}$: Nomic v1 (Nussbaum et al., 2025) | $0.27 \pm 0.02$ | $0.356 \pm 0.02$ | $\mathbf{0.136 \pm 0.03}$ | $0.446 \pm 0.04$ | $0.273 \pm 0.06$ | $0.417 \pm 0.04$ | $0.237 \pm 0.03$ | $0.336 \pm 0.03$ |
| $\mathcal{I}$: Qwen 2 (7B) (Yang et al., 2024a) | $0.0811 \pm 0.02$ | $0.268 \pm 0.03$ | $0.0203 \pm 0.01$ | $0.488 \pm 0.04$ | $\mathbf{0.355 \pm 0.06}$ | $0.42 \pm 0.04$ | $0.124 \pm 0.03$ | $0.299 \pm 0.03$ |
| $\mathcal{I}$: SF Mistral (Meng et al., 2024) | $0.0842 \pm 0.02$ | $0.296 \pm 0.03$ | $0.0128 \pm 0.01$ | $0.446 \pm 0.04$ | $0.29 \pm 0.06$ | $0.46 \pm 0.04$ | $0.105 \pm 0.02$ | $0.237 \pm 0.03$ |
| $\mathcal{I}$: Mistral (E5) (Wang et al., 2024a) | $0.112 \pm 0.02$ | $0.318 \pm 0.02$ | $0.00276 \pm 0.004$ | $0.445 \pm 0.04$ | $0.281 \pm 0.06$ | $\mathbf{0.471 \pm 0.04}$ | $0.0872 \pm 0.02$ | $0.228 \pm 0.03$ |
| $\mathcal{I}$: Cohere v4.0[†] (AI, 2025) | $0.11 \pm 0.02$ | $0.277 \pm 0.03$ | $0.0035 \pm 0.004$ | $\mathbf{0.504 \pm 0.03}$ | $0.299 \pm 0.07$ | $0.408 \pm 0.04$ | $0.239 \pm 0.03$ | $0.408 \pm 0.03$ |
| $\mathcal{I}$: Cohere v3.0[†] (AI, 2023) | $0.111 \pm 0.02$ | $0.277 \pm 0.03$ | $0.00325 \pm 0.004$ | $0.502 \pm 0.04$ | $0.265 \pm 0.07$ | $0.408 \pm 0.04$ | $0.198 \pm 0.03$ | $0.343 \pm 0.03$ |
| $\mathcal{I}$: Cohere v3.0 (Lite)[†] (AI, 2023) | $0.11 \pm 0.02$ | $0.278 \pm 0.03$ | $0.00324 \pm 0.004$ | $0.502 \pm 0.03$ | $0.236 \pm 0.06$ | $0.41 \pm 0.04$ | $0.247 \pm 0.03$ | $0.36 \pm 0.03$ |
| $\mathcal{I}$: Qwen 3 (0.6B) (Yang et al., 2025) | $0.0477 \pm 0.02$ | $0.172 \pm 0.02$ | $0.0147 \pm 0.01$ | $0.466 \pm 0.04$ | $0.305 \pm 0.06$ | $0.421 \pm 0.04$ | $0.182 \pm 0.03$ | $0.395 \pm 0.03$ |
| $\mathcal{I}$: Qwen 3 (4B) (Yang et al., 2025) | $0.00524 \pm 0.006$ | $0.165 \pm 0.03$ | $0.135 \pm 0.03$ | $0.441 \pm 0.04$ | $0.307 \pm 0.06$ | $0.368 \pm 0.05$ | $0.306 \pm 0.03$ | $0.395 \pm 0.03$ |
| $\mathcal{I}$: Qwen 3 (8B) (Yang et al., 2025) | $0.00247 \pm 0.004$ | $0.116 \pm 0.02$ | $0.126 \pm 0.03$ | $0.432 \pm 0.04$ | $0.279 \pm 0.06$ | $0.331 \pm 0.05$ | $\mathbf{0.329 \pm 0.03}$ | $0.361 \pm 0.04$ |

**Table 4: Main Experimental Results: Segment-Score**. Comparison of semantic isotropy with other uncertainty metrics across **TriviaQA Entities**, **FactScore-Bio** and **Book Summaries** datasets using the *Segment-Score* algorithm. Each dataset column is subdivided by base model: Llama 3.1 8B, Phi 3.5 Mini, and OpenAI 4.1 Mini (except *Book Summaries*). Values are $R^2$ (explained variance) of a simple linear model of Factuality $\sim$ Isotropy score. 1500 Bootstrapped samples are used to generate 1-SD error bars. †denotes API-only models. Note that Eigenscore values are not generated for GPT 4.1 Mini as it is a black-box model and the internal layer activations are not available.

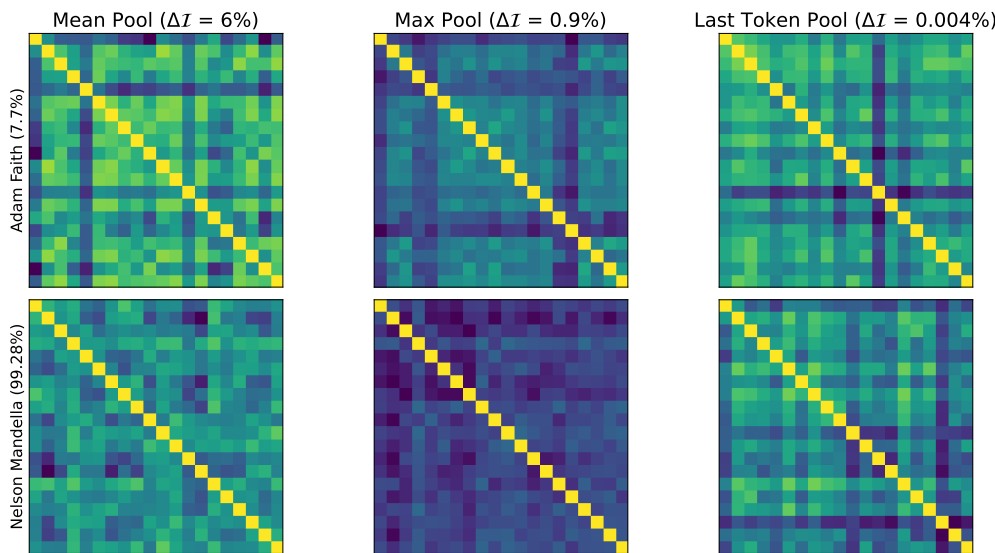

**Figure 6: Nomic v1 Embedding Choice**. $K_E^{cos}$ for two sample topics: **Adam Faith** (low factuality, top row) and **Nelson Mandela** (high factuality, bottom row). The number in the parenthesis next to the topic name is the average factuality across sampled responses. $\Delta\mathcal{I}$ for each pooling method denotes the percentage difference in semantic isotropy scores between the two topics (as a percentage of $\mathcal{I}$ for **Nelson Mandela**). We infer that Mean Pooling for Nomic V1 yields the most informative matrices for an isotropy analysis.

Embedding v4.0, v3.0 (AI, 2023; 2025) were also used to compute response embeddings for semantic isotropy computation. Additionally, OpenAI GPT 4.1, Anthropic Claude 4 Sonnet, and DeepSeek (DeepSeek-AI et al., 2024) were also used in verification runs to study robustness of the scoring pipeline. Overall, we utilize approximately 1,440 GPU hours (in addition to CPU compute time) to generate samples and score the responses.

## B.2 DEPLOYING SEMANTIC ISOTROPY SCORE IN A TASK-SPECIFIC CONTEXT

It is important to understand the practical interpretation of the *semantic isotropy* score, deployed in an actionable context. As measures of utility or correctness over long-form generations could be real valued and non-binary, we first discretize our experimental setup. We use the factuality score obtained by *Segment-Score* to classify all responses with a average factuality of $> 50\%$ as 1, and 0 otherwise. Using the $1 - \mathcal{I}$ as our confidence measure,

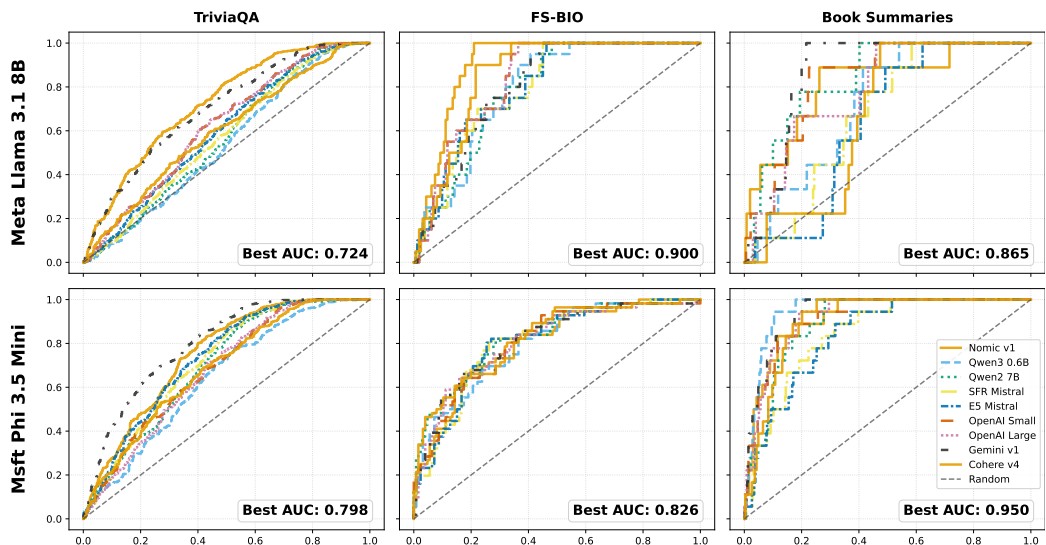

**Figure 7: Isotropy vs Factuality**. Receiver-Operator Characteristic curves comparing *Semantic Isotropy* to 0 / 1 classes obtained by discretizing *Segment-Score* by $\geq 50\%$ accuracy. Curves generated on all responses by Llama 3.1 and Phi 3.5 mini, on all 3 datasets (**TriviaQA, FS-BIO** and **BS**). In general we find our results are consistent with earlier results whereby semantic isotropy is predictive of nonfactuality.

Figure 7 outlines the Receiver-Operator characteristics for our *semantic isotropy* metric. The plots are consistent with our earlier results, with performance varying across the datasets, yet working well in most settings. In the more specific datasets, such as **FS-BIO** and **BS**, the score performs remarkably well at classifying responses.

### B.3 SEGMENT-SCORE IMPLEMENTATION SPECIFICS

Unlike *FactScore*, the *Segment-Score* algorithm is achieved through a single prompt per response. The basic prompt template for our method can be found in Figure 12. The prompt is structured into three components. First, an instruction block that covers the overall objective of the scoring exercise, expected data, desired output structure, and specific handling for edge cases. Next, we provide two manually curated examples to guide the model's output. One such example can be found in Figure 13. Last, we provide the entity name, contents of the reference document and response to be scored. The model is prompted to respond using a XML tag structure which simplifies the data extraction process. We also force the model to respond with only '0' or '1' for the output classes. For 99.8% of segments, the top 2 tokens are the '0' or '1' class labels, and we can use the log-probabilities from the API to generate a normalized probability score for each labeled segment.

## C ABLATION STUDIES

### C.1 RESPONSE GENERATION USING OPENAI GPT 4.1 MINI

To understand the impact of the generator model on the performance of the semantic isotropy score, we also generate responses using OpenAI GPT 4.1 Mini (OpenAI, 2023), scored using *Segment-Score*. Figure 8 shows the results of this experiment in the same format as our main results. For the **FS-BIO** dataset, we see semantic isotropy remains predictive of overall response factuality. However, we see a notable drop in performance in the case of **TriviaQA**. Notably, this extends to all benchmark metrics as well. We also observe much greater variance in the performance across embedding models, suggesting there are some settings where the choice of embedding model is crucial.

Consequently, it was important to understand the impact of the generation model on the performance of the semantic isotropy score. Zhang et al. (2024) note that the LUQ method does not perform well using GPT 4, a fact we were able to replicate.

> *"We also observe that LUQ is better suited for models with relatively lower factuality and a lack of self-expressiveness regarding uncertainty. For models with high factuality*

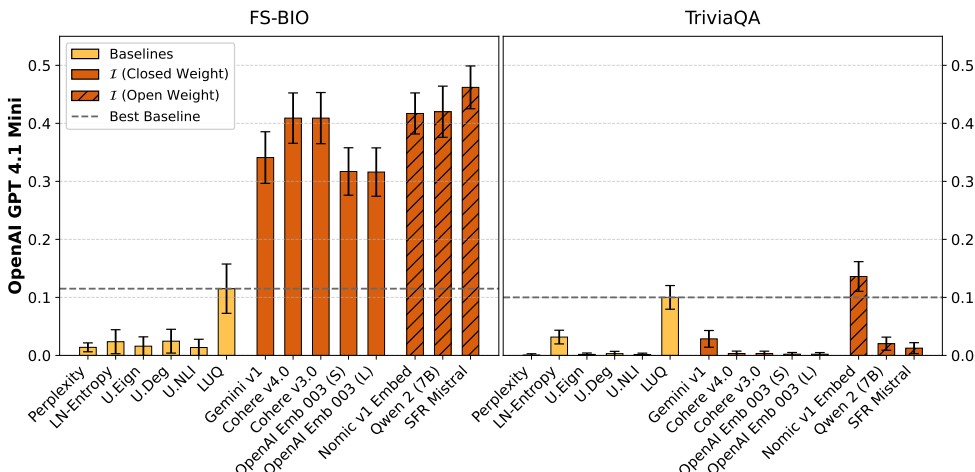

**Figure 8: Experimental Results using OpenAI GPT 4.1 Mini**. Bar Charts comparing the performance (as measured by the $R^2$ of a linear model of factuality with semantic isotropy as the explanatory variable; implemented using various embedding models and benchmark uncertainty metrics on both the **FS-BIO** ant **TriviaQA** datasets using the *Segment-Score* (**SS**) scoring algorithm. Left hand side: Benchmark UQ metrics. Right Hand Side: semantic isotropy ($\mathcal{I}$) implemented using various embedding models. Same experimental setting as in Figure 2. We observe that performance on generations using OpenAI GPT 4.1 Mini is significantly lower than other generation models, however this extends to all benchmark metrics as well. Semantic isotropy continues to outperform all other baselines. Note that Eigenscore values are not generated for GPT 4.1 Mini as it is a black-box model and the internal layer activations are not available.

> *capabilities, such as GPT-4, LUQ only demonstrates a moderate correlation with factuality scores"* - Zhang et al. (2024)

One would expect that GPT 4.1 Mini would perform better in terms of instruction following and maintaining consistent structure. This is anecdotally observed in the token count distribution (see Figure 10b), where GPT 4.1 Mini adhere's much better to the 500 word guidance given in the prompt across different topic areas, as compared to the other generative models being considered.

When studying the distribution of semantic isotropy scores, in Figure 10a, we observe that OpenAI GPT 4.1 Mini's scores are smaller in magnitude and more tightly clustered. This highlighted that there may be some inherent structural properties of the OpenAI GPT 4.1 Mini generations that manifested in the embeddings in this specific way. To tackle this, we performed a comparative study between generations from GPT 4.1 Mini (OpenAI, 2023), Phi 3.5 Mini (Abdin et al., 2024) and Llama 3.1 8B (Touvron et al., 2023). Using a LLM as a judge setup, we asked OpenAI GPT 4.1 to compare and contrast 10 generations from GPT 4.1 Mini vs 10 from either Phi 3.5 Mini or Llama 3.1 8B for 50 different topics where the all the models had relatively low factuality (i.e. $\leq 33\%$ accuracy on average). The LLM generated comparisons were summarized into a brief analysis (See Figure 9). This highlighted that properties such as structure, tone and other qualitative aspects of a generation materially impacted the resulting embeddings. We expect that improvements in embedding models or models specifically tuned to disregard qualitative aspects of a generation could improve the performance of the semantic isotropy score, but leave this exploration for future work.

## C.2 COMPARING THE EFFICIENCY OF THE TWO SCORING METHODS

*Segment-Score's* overall process closely follows the FactScore algorithm. However, we make several key optimizations that improve runtime. First, unlike FactScore, we break the input text into atomic segments rather than extracting atomic facts. As a result, each sentence may be broken into at most 2-3 segments that need to be scored whereas FactScore may generate several atomic facts pertaining to each sentence segment. For **FS-BIO**, this translates to $28 \pm 6$ distinct segments for each response in *Segment-Score*, while *FactScore* produces $100 \pm 20$ atomic facts (2-$\sigma$ interval).

Second, FactScore does not distinguish between verifiable and ambiguous facts which increases noise in the final results. We explicitly prompt the model to classify such examples as False. This generally biases our result lower, which can be seen in the middle plot of Figure 11.

Across the comparisons, Model A consistently demonstrates formal, polished, and
    structured writing with high factual accuracy, depth, and analytical rigor.
    Its responses are typically essay-like, varied in perspective, and nuanced,
    often employing advanced vocabulary, rhetorical devices, and critical
    engagement with broader themes such as legacy, cultural context, and impact.

In contrast, Model B tends to be more informal, conversational, and formulaic,
    frequently exhibiting repetition, superficial coverage, and factual errors.
    Model B's diversity often arises from inconsistency or error rather than true
    creativity, and it outputs are generally less coherent, less organized, and
    less engaging than Model A's.

**Figure 9:** Summary Analysis of Low Factuality Generations by GPT 4.1 Mini (Model A) vs Llama 3.1 8B (Model B).

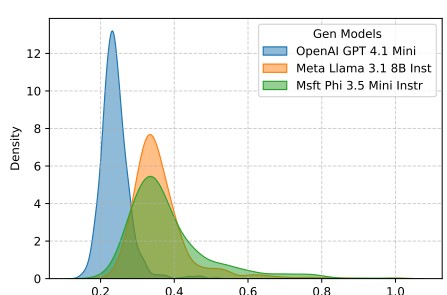

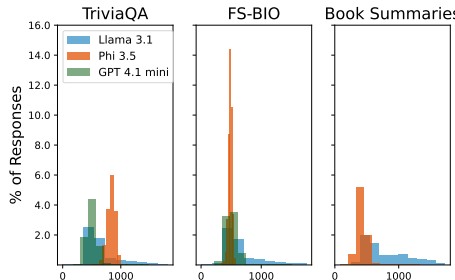

(a) **Kernel Desity Estimation plots of semantic isotropy scores for TriviaQA**. Kernel Density Estimation plots of semantic isotropy scores for TriviaQA, comparing the distributions of scores for GPT 4.1 Mini, Phi 3.5 Mini and Llama 3.1 8B. The embedding used in Gemini v001 (Team, 2024). We observe that GPT 4.1 Mini's scores are larger in magnitude and more tightly clustered.

(b) **Word Count by Generation Model and Dataset**: Density histograms comparing the response length (in tokens) for each generative model and dataset for a prompt with a guidance length of 500 words. We observe that GPT 4.1 Mini is much better at instruction following that Llama 3.1 and Phi 3.5 Mini, sticking fairly closely to the intended word limit.

**Figure 10: Response Characteristics:** Comparing the characteristics of generated responses by model type and dataset. Distribution of semantic isotropy scores for TriviaQA using Gemini [Left] and Histogram of response lengths in tokens by dataset and generative model [Right].

Finally, our entire process is achieved using one single query to the oracle, while FactScore may use an order of magnitude more depending on the number of atomic facts extracted and the length of the ground truth document. Consequently, our method scales more favorably as the length of the response to be scored increases. Despite these differences, we see comparable results for **FS-BIO** in Table 3 and Table 4 for each scoring method when considering the various isotropy configurations and response generating models.

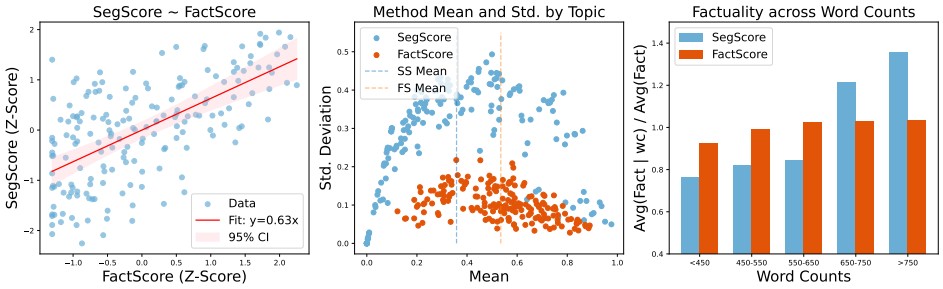

**Figure 11: FS vs SS [FS-BIO]**. Comparison of *FactScore* and *Segment-Score* on the **FS-BIO** dataset. *Segment-Score* yields a dataset with lower overall factuality and higher variance, however the relationship between the two scores is robust and consistent.

## D    LIMITATIONS

The first drawback of our method is the need for repeated sampling, which can be expensive to generate. Approaches such as Kossen et al. (2024) have shown that it is possible to train cheap probes that can a priori

estimate uncertainty scores without needing repeated sampling. With the advent of reasoning models with long Chain-of-Thought reasoning contexts (Zhao et al., 2024) and Monte-Carlo Tree Search (Wang et al., 2024b), as well as fast inference engines with significantly higher throughput (Kwon et al., 2023), the relative overhead of repeated sampling has been greatly diminished, making methods like ours compelling.

Additionally, while studies such as Manakul et al. (2023) have observed that hallucinated responses tend to diverge, and generally are inconsistent with each other, Ricco et al. (2025) find that this effect is not universal and introduce the taxonomy of *extrinsic* (characterized by arbitrary and orthogonal content) vs *intrinsic* (contradictory or anti-parallel content) hallucination. The semantic isotropy framework would likely fail in cases of *intrinsic* hallucinations. However, in practice this intrinsic hallucinations would be exceedingly rare, especially in the long-form response format.

Finally, it is important to call out that our method is not designed to work in adversarial scenarios. In the event that one were to use *semantic isotropy* on a model that had been intentionally trained on faulty content, semantic consistency through samples would only indicate the model is faithfully inferring from its training corpora, and would not indicate factuality or trustworthiness. While *semantic isotropy* remains far more robust to naive manipulation techniques such as ballot stuffing, whereby facts are repeatedly stated to boost accuracy metrics, this failure case is beyond the scope of standard embedding models. In this case, we deem *semantic isotropy* as a complementary early-warning signal, rather than a full fledged replacement for claim-by-claim verification.

```
// INSTRUCTION PROMPT FOR SEGMENT SCORE
You are an NLP segmentation and evaluation engine.
Examine the scenario below. You are given:
1. The name of an entity/person/place/thing etc. in <entity> tags.
2. A reference document regarding the entity in <reference_doc> tags.
3. A response about the entity to evaluate in <response> tags.

### Your Tasks:
1. **Segmentation Task:**
   Segment the `<response>` into individual statements. Each statement can be a sentence,
       phrase or word and should convey a single, complete, and independent piece of
       information about the `<entity>`. Do not modify, rephrase, or paraphrase the
       original text. Ensure no semantic overlaps exist between statements. Individual
       proper nouns should be part of their own statement however when determining the
       appropriate classification, preceeding context can be used when appropriate.
   Verify that the concatenated content of all statements exactly matches the original
       response.

   Format the segmented response as follows:
   ```
   <statements>
   <statement>Statement 1</statement> <class>Class 1</class>
   <statement>Statement 2</statement> <class>Class 2</class>
   ...
   </statements>
   ```

2. **Factual Classification Task:**
   For each segmented `<statement>`, classify it as 1 (True) or 0 (False) based solely
       on the information in the `<reference_doc>`. Follow these guidelines:
   - If a statement is factually accurate and supported by the `<reference_doc>`,
       classify it as '1'.
   - If a statement is inaccurate, unverifiable, or not supported by the `<reference_doc
       >`, classify it as '0'.
   - If a statement is partially true, but contains incorrect or unsupported information,
        classify it as '0'.
   - Do not rely on any external knowledge or context beyond the `<reference_doc>`.
   - Include only the classification for each statement. Do not provide any explanations
       or additional information.
   - Specify the class in <class> tags.
   - The ONLY valid class values are `1` and `0`. No other values or words should appear
       within the `<class>` tags.

3. **Error Handling:**
   - If the `<response>` contains unparseable text, incomplete sentences, or conflicting
        information that cannot be resolved using the `<reference_doc>`, include the
       flagged statement as is and classify it as 'False'.

Examples:
####### EXAMPLE 1 ######
{{example_one}}
#######################
####### EXAMPLE 2 ######
{{example_two}}
#######################

Entity:
<entity>
{entity}
</entity>

Reference Document:
<reference_doc>
{reference_doc}
</reference_doc>

Response to Evaluate:
<response>
{response}
```

```
<entity>
London, UK
</entity>

Reference Document:
<reference_doc>
London, England's capital, boasts a rich history spanning millennia. Founded by the
    Romans as Londinium around 47 AD, it became a major port and trading center. After
    the Roman withdrawal, Anglo-Saxons established Lundenwic, which later fell to
    Viking raids. The Norman Conquest in 1066 led to the construction of the Tower of
    London, a symbol of royal power. London thrived during the medieval period,
    becoming a major center for trade, finance, and culture. It weathered plagues,
    fires, and civil wars, emerging as a global metropolis and the heart of the British
    Empire. Today, London remains a vibrant hub, blending its historical legacy with
    modern dynamism, home to over 9 million people.
</reference_doc>

Response to Evaluate:
<response>
London, the capital city of England and the United Kingdom, is a vibrant metropolis
    steeped in history and brimming with modern energy. With a population of over 9
    million people, it stands as one of the world's most influential global cities,
    known for its diverse culture, iconic landmarks, and rich heritage.

The city's history stretches back over three millennia, founded by the Romans as
    Londinium in 43 AD. Throughout the centuries, London has played a pivotal role in
    world affairs, serving as the heart of the British Empire and surviving tumultuous
    events such as the Great Fire of 1666 and the Blitz during World War I.

Today, London is a melting pot of cultures, with over 300 languages spoken within its
    boundaries. This diversity is reflected in its neighborhoods, each with its own
    unique character and charm. From the trendy streets of Shoreditch to the upscale
    boutiques of Mayfair, there's something for everyone in this cosmopolitan city.
</response>

Segmented and classified response:
<statements>
<statement>London, the capital city of England and the United Kingdom</statement> <class
    >1</class>
<statement>is a vibrant metropolis steeped in history and brimming with modern energy</
    statement> <class>1</class>
<statement>With a population of over 9 million people</statement> <class>1</class>
<statement>it stands as one of the world's most influential global cities, known for its
     diverse culture, iconic landmarks, and rich heritage.</statement> <class>1</class>
<statement>The city's history stretches back over three millennia, founded by the Romans
     as Londinium in 43 AD</statement> <class>0</class>
<statement>Throughout the centuries, London has played a pivotal role in world affairs,
    serving as the heart of the British Empire</statement> <class>1</class>
<statement>and surviving tumultuous events such as the Great Fire of 1666</statement> <
    class>1</class>
<statement>and the Blitz during World War I</statement> <class>0</class>
<statement>Today, London is a melting pot of cultures, with over 300 languages spoken
    within its boundaries</statement> <class>1</class>
<statement>This diversity is reflected in its neighborhoods, each with its own unique
    character and charm.</statement> <class>1</class>
<statement>From the trendy streets of Shoreditch to the upscale boutiques of Mayfair,
    there's something for everyone in this cosmopolitan city</statement> <class>1</
    class>
</statements>
```

**Figure 13:** Segment-Score Example used as part of model in-context training for oracle LLM.

