# OpenReview forum: "Embedding Trust: Semantic Isotropy Predicts Nonfactuality in Long-Form Text Generation"
_ICLR.cc/2026/Conference — Submitted to ICLR 2026_

### Official Review · Reviewer_esvS · 2025-10-31

**Soundness:** 2
**Presentation:** 3
**Contribution:** 2
**Rating:** 4
**Confidence:** 4

**Summary:**

This paper introduces a computationally efficient, embedding-based approach to approximating the factuality of LLM responses. The Semantic Isotropy approach samples multiple answers, runs them through embedding models, and then estimates the von Neumann entropy of the cosine kernel under the set of embeddings. This approximator is evaluated via its correlation with SegmentScore, a coarse-grained version of FactScore that this paper also introduces. The approximator outperforms numerous baselines under the correlation metric using numerous models for generating and embedding answers. The paper contributes the estimator, the new metric, and a new dataset of LLM-generated long form answers over subsets of FS-BIO and TriviaQA.

**Strengths:**

S1. The method's computational efficiency would be highly beneficial if the experimental evidence gives strong support for using it in a downstream task.

S2. The authors comprehensively evaluate multiple baseline approximators, answer generators, and embedding models.

S3. The paper is well-written with no obvious methodological details missing.

**Weaknesses:**

W1. The approach is evaluated using a new metric, Segment-Score, that is not validated for its coherence. FactScore, on which the new metric is based, was evaluated against ground-truth human annotations to verify its correlation. Especially since Segment-Score is less granular than FactScore (evaluating at the level of segment rather than atomic fact) and has higher variance, it is important to verify that it still has a low ground truth error rate. The authors show a reasonable ($R^2 = .63$) correlation, but this is not perfect and merits further validation.

W2. The approach requires overgenerating many responses (8+) to get a nontrivial signal of embedding isotropy, so cannot be used off-the-shelf to evaluate one given response.

W3. The overall $R^2$ numbers for certain configurations, e.g. Meta Llama on TriviaQA, are quite low; while the approach outperforms the considered baselines as an approximation for the factuality score, it raises the question of if any of these approaches are at all useable if none of them achieve a certain threshold of calibration.

W4. In general, the paper treats $R^2$ as an end goal for an approximator, but that doesn't tell us if the method will be practically useful.

**Questions:**

Q1. L306 "we exclude all entities that correspond to days, dates or numerical values and only select those that match the title of the underlying Wikipedia page." -- the motivation for this is unclear, why did you perform this subselection?

Q2. Does the approach work for cases in which the LLM response is more sparse in its factual statements? e.g. One response might contain [Fact1, Fact2] while another might contain [Fact2, <elaboration of fact2>, Fact1, <elaboration of fact1>] -- i would imagine the embedding of these two responses might be further from each other than [Fact1, IncorrectFact3].

Q3. Why is SegScore higher on average for >750 length responses than it is on <650?

Q4. How might this approach be put into practice? Does the method actually work for detecting factual problems?  It would be helpful to perform task-specific validation of some sort -- using the method as a verifier with a task-specific success criterion (e.g. how about using the FactScore itself to evaluate responses and then compute an ROC curve across various thresholds of Semantic Isotropy?)

Q5. The approach violates my intuitions about LFQA. Multiple 500+ token responses that are both factual but, perhaps, say different things about a subject (e.g. one response "tell me about Marie Curie" is about where Marie Curie lived and another is about her research; -- semantically very different, so likely different embeddings). Wouldn't this approximator penalize this sort of behavior?

---

> ### Author Response · Authors · 2025-11-25
> **Author Response (1/5)**
>
> Thank you for your careful reading of our manuscript and for your detailed questions. We address your questions and concerns below, with particular attention to the methodological foundations and practical applicability of the proposed approach.
>
> ---
>
> > W1. The approach is evaluated using a new metric, Segment-Score, that is not validated for its coherence. FactScore, on which the new metric is based, was evaluated against ground-truth human annotations to verify its correlation. Especially since Segment-Score is less granular than FactScore (evaluating at the level of segment rather than atomic fact) and has higher variance, it is important to verify that it still has a low ground truth error rate. The authors show a reasonable ($R^2 = 0.63$) correlation, but this is not perfect and merits further validation.
>
> We agree that full human annotation would provide the most rigorous validation. Unfortunately, such an evaluation is not feasible within the time frame of this exercise. To demonstrate that Segment-Score is indeed consistent and coherent, we performed an extensive multi-model consistency evaluation using several strong verifiers (GPT-4-class models, Claude, Gemini and DeepSeek). We sampled a subset of our dataset from FS-BIO and BookSummaries (30 entities = 300 responses from each. We then asked the frontier model from OpenAI, Claude and DeepSeek to classify the statements generated for each response from Segment-Score against the ground truth. We took 5 samples of the classification and used majority vote to come up with reference labels. We then check alignment between these labels and the labels one-shot labels generated using Segment-Score.
>
> Results are shown in the table below:
>
> | Dataset / Generative Model | OpenAI (GPT 4.1) | Claude (4 Sonnet) | DeepSeek |
> | :--- | :--- | :--- | :--- |
> | **BookSummaries (Llama 3.1)** | 89.24 | 81.33 | 75.87 |
> | **BookSummaries (Phi 3.5 Mini)** | 91.28 | 86.13 | 80.38 |
> | **FS-BIO (Llama 3.1)** | 83.92 | 86.13 | 86.39 |
> | **FS-BIO (Phi 3.5 Mini)** | 76.98 | 86.67 | 88.30 |
>
> While there is variability, we note that in most cases the alignment is high. In the case of FS-BIO on Phi 3.5 Mini, given the small sample size we can attribute some of the drop in agreement to sampling bias.
>
> Additionally, **as you noted, we obtained a correlation coefficient of 0.63 with FactScore on the same responses (Figure 10)**. While not perfect, this correlation is substantial given that the two methods use different granularities (segments vs. atomic facts) and different verification procedures. **The systematic relationship between the two scores suggests Segment-Score captures similar underlying factual properties.**

---

> ### Author Response · Authors · 2025-11-25
> **Author Response (2/5)**
>
> > W2. The approach requires overgenerating many responses (8+) to get a nontrivial signal of embedding isotropy, so cannot be used off-the-shelf to evaluate one given response.
>
> There may be a misunderstanding. While our method does requires at least two samples to estimate dispersion, we find that as few as few as two samples are enough to obtain a useful signal. That being said, even a higher sampling burden--which we showed can impove the usefulness of isotropy scoring--is still practical for several reasons:
>
> - First, sampling efficiency has improved dramatically. **Modern inference engines like vLLM (Kwon et al., 2023) support highly efficient parallel sampling**, allowing 10-20 responses to be generated nearly as quickly as a single response through batching and KV-cache reuse.
> - Second, **many real-world applications naturally involve multiple generations**. Retrieval-augmented generation systems often generate multiple candidate responses for re-ranking. Chatbot deployments frequently use ensemble methods or majority voting, both of which require multiple samples.
> - Third, **the sample requirement (6-10 responses) is modest compared to the computational cost of alternatives**, such as running a "LLM-as-a-judge" setup for verification.
> - Last, **a number related works in this area also use repeated sampling for their methods.** Semantic Entropy Probes (Kossen et al. 2024) builds on Semantic Entropy by providing a algorithm that does not require repeated sampling. The results are impressive but are significantly inferior to using Semantic Entropy directly. This represents an inherrent trade-off in performance vs accuracy that should rightly sit with the end user desinging the specific application where this method is being used.
>
> ---
>
> > W3. The overall $R^2$ numbers for certain configurations, e.g. Meta Llama on TriviaQA, are quite low; while the approach outperforms the considered baselines as an approximation for the factuality score, it raises the question of if any of these approaches are at all useable if none of them achieve a certain threshold of calibration.
>
> We agree that this problem is fundamentally difficult and that all existing methods struggle in this setting, including FactScore and the baselines we compare against. However, we do not claim that the problem is fully solved and instead to show--with substantial empirical evidence--that **semantic isotropy offers a meaningful and efficient signal that consistently outperforms competing methods.**
>
> We agree that calibration remains an open challenge and have strengthened our discussion of when and why performance varies (Section 5, Appendix C.1). However, our method consistently achieves state-of-the-art performance compared to existing baselines (LUQ, EigenScore, etc.). **Providing the best available signal—which outperforms current standards by a wide margin—is a significant step forward for lightweight monitoring.**

---

> ### Author Response · Authors · 2025-11-25
> **Author Response (3/5)**
>
> > W4. In general, the paper treats $R^2$ as an end goal for an approximator, but that doesn't tell us if the method will be practically useful.
>
> You're raising a deeper point about the evaluation of language models, and we agree that downstream task usefulness is difficult to gauge. That being said, under the reasonable assumption that having a score (e.g., our semantic isotropy score) that provides a signal as to whether or not to trust an LLM's long-form predictions for a given topic is useful, showing that a given signal (e.g., our semantic isotropy score) is highly predictive of the variation in ground truth trust scores (which is what a high $R^2$ value tells us) seems to be a potentially good proxy for practical usefulness.
>
> That being said, the utility of our method of course depends on the specific deployment context. For applications requiring absolute factual guarantees, no uncertainty quantification method (including ours) replaces gold-standard verification--however, **our method reduces verification cost by up to 150$\times$ through efficient filtering**. For applications with softer requirements (content moderation, quality monitoring, user warnings), even moderate correlation provides actionable signals.
>
> ---
>
> > Q1. L306 "we exclude all entities that correspond to days, dates or numerical values and only select those that match the title of the underlying Wikipedia page." -- the motivation for this is unclear, why did you perform this subselection?
>
> We performed this filtering to mitigate the "Aboutness" problem. Numerical entities (e.g., "1990") are inherently vague as prompt subjects. Asking a model to "Write about 1990" could yield valid factual responses about politics, music, or sports that appear semantically diverse (high isotropy) despite being factual. This would confound our metric. We filtered for specific named entities to ensure that a "factually consistent" response has a well-defined semantic target.
>
> For example, a TriviaQA question might be "When did the 2004 Indian Ocean Tsunami occur?" with answer "December 26, 2004." Converting this to an open-ended long-form prompt ("Write paragraphs about 'December 26, 2004'") produces either a vague calendar description or forces the model to infer the intended topic (the tsunami). Neither case provides a well-defined factual target.
>
> Furthermore, reference document mismatch creates systematic evaluation errors. If we prompt "Write about Indonesia" (because Indonesia was the answer to a tsunami-related question), but the reference document is specifically about the 2004 tsunami event, then factually accurate content about Indonesian culture, history, or geography would be incorrectly marked as false because it's not covered in the tsunami-specific reference.
>
> Our filtering ensures that: (1) each entity represents a coherent topic admitting substantive long-form discussion, and (2) the Wikipedia reference document actually covers the intended topic scope. This alignment is critical for valid factuality assessment.
>
> The filtering yielded 5,245 TriviaQA entities across the training and validation sets, of which we randomly sampled 1,000 entities to generate long-form responses for. We have clarified this rationale in Section 4.1 of the manuscript.
>
> ---
>
> > Q2. Does the approach work for cases in which the LLM response is more  sparse in its factual statements? e.g. One response might contain [Fact1, Fact2] while another might contain [Fact2, [object Object], Fact1, [object Object]] -- i would imagine the embedding of these two responses might be further from each other than [Fact1, IncorrectFact3].
>
> Empirically, embedding-based covariance structures show strong robustness to variation in ordering, stylistic phrasing, and repeated content, as they are often trained on bi-directional attention architectures without the limitation of the causal mask.
> While we cannot address every sparsity case without specific data, we found that our method is generally robust to variations in content density. In early calibration experiments which included adversarial transformations such as ballot stuffing, and we observed minimal effect on embedding vectors, and thereby the semantic isotropy score.

---

> ### Author Response · Authors · 2025-11-25
> **Author Response (4/5)**
>
> > Q3. Why is SegScore higher on average for >750 length responses than it is on <650?
>
> We hypothesize two reasons for this trend.
> - First, for any given topic, there exists a set of highly salient facts that are most likely to be recalled correctly (e.g., for Marie Curie: her nationality, Nobel Prizes, field of research). Shorter responses may only sample a subset of recallable facts due to space constraints. As responses lengthen, they tend to cover more of these core facts, improving the true-positive rate.
> - The second effect is that longer responses often include more contextual and elaborative content—descriptions of historical significance, comparisons, narrative framing—that is easily verified against the reference without making specific novel claims. For instance, statements like "her groundbreaking work transformed scientific understanding" or "she remains an inspiration to scientists worldwide" are general enough to be supported by any comprehensive biography, while adding to response length. A third effect is *selective verbosity*. Models may be more verbose precisely when they have confident, well-grounded knowledge about a topic. When uncertain, models might produce terse responses. This creates a correlation between length and factuality that partially explains the trend.
>
> ---
>
> > Q4. How might this approach be put into practice? Does the method actually work for detecting factual problems? It would be helpful to perform task-specific validation of some sort -- using the method as a verifier with a task-specific success criterion (e.g. how about using the FactScore itself to evaluate responses and then compute an ROC curve across various thresholds of Semantic Isotropy?)
>
> We appreciate your concern regarding practical deployment. As we mentioned above, the utility derived over a long-from generation is highly context-dependent. In our case, *nonfactuality* is a continuous score over a set of generations. To view this through the lens of a factuality discriminant as you suggest, we discretized the FactScore and Segment-Score at 0.5 (50\% accurate). We show our results in the table below, with each entry corresponding to the AUROC obtained when comparing the semantic isotropy score with the binary factuality classification.
>
> | Metric | **Llama 3.1** BookSummaries | **Llama 3.1** TriviaQA | **Llama 3.1** FS-BIO | **Phi 3.5** BookSummaries | **Phi 3.5** TriviaQA | **Phi 3.5** FS-BIO | **OpenAI GPT 4.1** TriviaQA | **OpenAI GPT 4.1** FS-BIO |
> | :--- | :--- | :--- | :--- | :--- | :--- | :--- | :--- | :--- |
> | **perplexity** | 0.792284 | 0.792284 | 0.641333 | 0.679989 | 0.679989 | 0.776267 | 0.662075 | 0.600963 |
> | **length_norm_plogp** | 0.781790 | 0.781790 | 0.639003 | 0.678430 | 0.678430 | 0.776185 | 0.593707 | 0.586601 |
> | **U.eig** | 0.784259 | 0.784259 | 0.593022 | 0.386905 | 0.386905 | 0.687352 | 0.469898 | 0.481152 |
> | **U.deg** | 0.778395 | 0.778395 | 0.584207 | 0.385062 | 0.385062 | 0.678849 | 0.470408 | 0.476818 |
> | **U.nli** | 0.200309 | 0.200309 | 0.384038 | 0.628118 | 0.628118 | 0.299543 | 0.486395 | 0.508395 |
> | **eigenscore** | 0.543827 | 0.543827 | 0.447472 | 0.446287 | 0.446287 | 0.503811 | NaN | NaN |
> | **gemini-v001** | 0.804938 | 0.804938 | 0.701560 | 0.812925 | 0.812925 | **0.798246** | 0.676871 | 0.565799 |
> | **cohere-v4.0** | 0.855247 | 0.855247 | 0.595748 | 0.821570 | 0.821570 | 0.686312 | 0.734184 | 0.487274 |
> | **openai-v3-small** | 0.830864 | 0.830864 | 0.620475 | 0.809099 | 0.809099 | 0.685879 | 0.675000 | 0.481069 |
> | **openai-v3-large** | 0.830864 | 0.830864 | 0.620556 | 0.810941 | 0.810941 | 0.675541 | 0.674660 | 0.481139 |
> | **nomic_text_v1** | **0.900309** | **0.900309** | **0.724072** | 0.805272 | 0.805272 | 0.751163 | **0.793197** | **0.651721** |
> | **qwen3-4B** | 0.726852 | 0.726852 | 0.453639 | 0.807540 | 0.807540 | 0.632627 | 0.650000 | 0.306540 |
> | **qwen3-0.6B** | 0.791049 | 0.791049 | 0.538982 | 0.801587 | 0.801587 | 0.646287 | 0.682483 | 0.430259 |
> | **qwen2-7B** | 0.787654 | 0.787654 | 0.563956 | 0.825964 | **0.825964** | 0.699845 | 0.683673 | 0.414409 |
> | **sfr-mistral-7B** | 0.796296 | 0.796296 | 0.572427 | 0.809099 | 0.809099 | 0.716344 | 0.718367 | 0.430159 |
>
> [response continued below]

---

> ### Author Response · Authors · 2025-11-25
> **Author Response (5/5)**
>
> [continued from above]
>
> Additionally, inspired by your comment, we also evaluated our method in the context of the C-Index or the *Concordance* index, which measures if a predictor is rank preserving. We show these results in the table below:
>
> |  | meta-llama-3.1-8B-instruct | openai_gpt-4.1-mini | msft_phi3.5-mini-instruct |
> |:---|:---|:---|:---|
> | perplexity        | 0.593349 | 0.585187 | 0.539868 |
> | length_norm_plogp | 0.589200 | 0.549899 | 0.538954 |
> | U.eig             | 0.672618 | 0.450898 | 0.429651 |
> | U.deg             | 0.663691 | 0.447834 | 0.429042 |
> | U.nli             | 0.390118 | 0.551430 | 0.579188 |
> | eigenscore        | 0.577760 | 0.000000 | 0.467935 |
> | gemini-v001       | 0.780048 | 0.685965 | 0.695013 |
> | cohere-v4.0       | 0.803998 | 0.718189 | 0.705194 |
> | openai-v3-small   | 0.776528 | 0.683882 | 0.697330 |
> | openai-v3-large   | 0.776402 | 0.683637 | 0.697025 |
> | nomic_text_v1     | **0.811919** | **0.754334** | 0.708181 |
> | qwen3-4B          | 0.757606 | 0.668933 | 0.704645 |
> | qwen3-0.6B        | 0.788471 | 0.702383 | 0.700378 |
> | qwen2-7B          | 0.774453 | 0.690927 | **0.711900** |
> | sfr-mistral-7B    | 0.767161 | 0.716167 | 0.693855 |
>
> **Both results indicate that despite the problem setting being challenging, semantic isotropy scoring continues to outperform all applicable baseline measures in our experimental setting.**
>
> ---
>
> > Q5. The approach violates my intuitions about LFQA. Multiple 500+ token responses that are both factual but, perhaps, say different things about a subject (e.g. one response "tell me about Marie Curie" is about where Marie Curie lived and another is about her research; -- semantically very different, so likely different embeddings). Wouldn't this approximator penalize this sort of behavior?
>
> This is a good point and a valid intuition regarding Semantic Equivalence vs. Factual Consistence. If a model outputs valid but disjoint facts (e.g., one response covers "Early Life" and another covers "Career"), the embeddings will separate, leading to a higher isotropy score (indicating uncertainty). However, in practice, we find that with $N=10$ samples and 500-1,000 word responses, models tend to cover the "modal" facts of the entity. The likelihood of generating 10 completely disjoint but true biographies is low. The samples usually overlap significantly in content if the model is confident. High dispersion usually indicates the model is "grasping" for facts (hallucinating), rather than confidently stating disjoint truths.
>
> ---
>
> We hope that our response has satisfactorily addressed your questions and concerns, and **we would be grateful if you would consider supporting our work being presented at the conference**. Thank you for your time and feedback!

---

> ### Comment · Reviewer_esvS · 2025-11-26
>
> > Asking a model to "Write about 1990" could yield valid factual responses about politics, music, or sports that appear semantically diverse (high isotropy) despite being factual. This would confound our metric. We filtered for specific named entities to ensure that a "factually consistent" response has a well-defined semantic target.
>
> > Our filtering ensures that: (1) each entity represents a coherent topic admitting substantive long-form discussion, and (2) the Wikipedia reference document actually covers the intended topic scope. This alignment is critical for valid factuality assessment.
>
> > in practice, we find that with samples and 500-1,000 word responses, models tend to cover the "modal" facts of the entity. The likelihood of generating 10 completely disjoint but true biographies is low. The samples usually overlap significantly in content if the model is confident. High dispersion usually indicates the model is "grasping" for facts (hallucinating), rather than confidently stating disjoint truths.
>
>
> You seem to be claiming that your method works because models don't actually exhibit the diversity that would break the method, while acknowledging that diversity-promoting prompts like "write about 1990" would, in fact, break the method. To me, the filtered evalaution still seems somewhat cherry-picked in comparison to real world questions. You claim that 500-1000 word biographies fall on the "desirable" side of the boundary between scenarios in which response diversity correlates with uncertainty and scenarios in which diversity is a natural byproduct of the task/question.
>
> The validity of the approach for general-purpose question answering seems to hinge in part on whether your claim that 500-1000 word responses naturally converge on 'modal facts' is actually true across diverse topics and models. It would strengthen the paper to provide empirical validation of the claim.
>
>
> > our method consistently achieves state-of-the-art performance compared to existing baselines (LUQ, EigenScore, etc.).
>
> The baselines considered in the evaluation are all variations on consistency measurement between the N responses. They therefore all have the same issue that they only work in the scenarios where N factual responses should all be semantically similar. Outperforming other consistency-based methods doesn't validate the consistency-as-factuality assumption itself.

---

> > ### Author Response · Authors · 2025-12-03
> > **Author Response to Official Comment by Reviewer esvS (1/2)**
> >
> > We thank the reviewer for their continued engagement. However, we must respectfully push back on the premise of these critiques which we believe conflate the methodology of measurement with the definition of the task, and effectively challenge the validity of the entire sub-field of consistency-based uncertainty quantification rather than the specific contributions of our work. We believe it is important to respond directly and clearly.
> >
> > ---
> >
> > >You seem to be claiming that your method works because models don't actually exhibit the diversity that would break the method, while acknowledging that diversity-promoting prompts like "write about 1990" would, in fact, break the method. To me, the filtered evalaution still seems somewhat cherry-picked in comparison to real world questions
> >
> > We respectfully but firmly disagree with the characterization that our evaluation is "cherry-picked." Our experimental design reflects careful consideration of a well-defined and practically important problem: detecting epistemic uncertainty in factual long-form generation. In experimental design, one must control variables to isolate the phenomenon under study. Our work targets epistemic uncertainty (which results in hallucinations) in cases where the model should know the answer but fails. Scenarios like "Write about 1990" introduce high aleatoric/irreducible uncertainty, where multiple distinct answers are equally valid. In such cases, high dispersion (high isotropy) is not a "break" or failure of our method; it is the correct and accurate signal that the model is not converging on a single semantic cluster.
> >
> > In fact, prompts like “Write about 1990” fall into a different category in which aleatoric variation—genuinely different, equally reasonable ways to answer the question—is a feature of the task. In such settings, any pure consistency-based score (ours or others’) will inevitably entangle epistemic and aleatoric uncertainty. We explicitly do not claim that semantic isotropy is a universal factuality estimator for arbitrary open-ended prompts; rather, we position our work as addressing a specific and practically important regime: long-form knowledge recall and synthesis where there exists a clear, document-level notion of “being about the right thing”.
> >
> > We believe that using factuality grounded in reference documents for knowledge retrieval falls on the "tractable" side of the boundary. This experiment can be repeated in a number of different contexts, such as providing medical feedback based on patient symptoms and then using *semantic isotropy* to judge the quality of the suggested diagnosis. However we are not aware of any off the shelf, scalable methods to score these responses for "medical utility and faithfulness" other than using expert human evaluators. This was not a viable option for us given several constrains, including time and budget. Our core hypothesis is deliberately scoped to prompts for which there is a relatively well-defined, factually grounded answer manifold—e.g., answers tied to a unique, concrete reference document.
> >
> > Finally, we note that our evaluation spans 1,691 unique entities across three diverse datasets, three generator models (including state-of-the-art GPT-4.1 Mini), and 13 embedding models. We evaluate biographical content, general knowledge, and literary summaries—achieving consistent results across all settings. If our method only worked on a narrow slice of carefully selected examples, seeing such a robust performance across a wide range of diverse conditions would be very unlikely.
> >
> > ---

---

> > > ### Author Response · Authors · 2025-12-03
> > > **Author Response to Official Comment by Reviewer esvS (2/2)**
> > >
> > > > The validity of the approach for general-purpose question answering seems to hinge in part on whether your claim that 500-1000 word responses naturally converge on 'modal facts' is actually true across diverse topics and models. It would strengthen the paper to provide empirical validation of the claim.
> > >
> > >
> > > It can be easily verified either through simulation or algebra, that the larger a sample from a set is, the sooner the union of many repeated samples converges to the full set. Within our problem setting, we have empirically validated that this is true across several topics and generative models.
> > >
> > > As noted above, what you are describing is not so much a characteristic of our work, or our method, than it is a property of all factuality / knowledge recall-style experiments where each query has a ground truth reference. By definition, such a reference would be limited and finite. Therefore the premise that you asked to be empirically validated is a property of our experimental setting that can be confirmed purely through deductive reasoning.
> > >
> > > Most importantly, we want to point out that we show in our manuscript that using *semantic isotropy* to predict nonfactulaity works perfectly well for shorter response lengths as it does for longer ones. This empirical finding is in direct contradiction with your stated premise that *semantic isotropy* would only work in 500-1000 word long responses that converge to the "modal facts". We show that our method achieves state-of-the-art desults even for responses as short as 125 words. We do believe in general that, as responses get longer, more of the salient facts are covered. But even at this short response length, *semantic isotropy* can detect when the LLM is converging to a well defined semantic cluster, as opposed to hallucinating random claims.
> > >
> > > ---
> > >
> > > > The baselines considered in the evaluation are all variations on consistency measurement between the N responses... Outperforming other consistency-based methods doesn't validate the consistency-as-factuality assumption itself.
> > >
> > >
> > > We agree that our comparisons are conducted within a family of methods sharing a common premise: that when tasks "should" admit relatively specific answers, cross-sample consistency can be informative about model reliability. Our contribution is expressly situated within this established line of work.
> > >
> > >
> > > To clarify, our paper makes two key claims:
> > >
> > > 1. Within consistency-based methods, semantic isotropy provides a more robust signal than prior approaches (NLI-based checks, log-prob proxies, EigenScore, LUQ, etc.). Our experiments consistently support this across datasets, generators, and embedding families.
> > >
> > > 2. There exists a practically important class of applications where consistency-based approaches are justified—specifically, when LLMs serve as knowledge recall and synthesis engines over well-covered topics. In these settings, hallucination and partial recall are the dominant failure modes, and consistency-based signals are both natural and operationally useful.
> > >
> > > We acknowledge in our limitations that consistency and factuality can diverge in adversarial scenarios. Our method does not attempt to solve those settings. Instead, we view semantic isotropy as a valuable tool wherever consistency-based premises are already adopted by practitioners—which encompasses substantial current work in alignment and reliability.
> > >
> > > We appreciate your engagement with these foundational questions. While we may not fully agree on the philosophical scope of consistency-based methods, we hope we have demonstrated that within this established and practically important paradigm, semantic isotropy represents a meaningful advance. We believe this contribution—achieving state-of-the-art performance with 150× speedup, no training data, and robust generalization—will be of significant interest to the safety and alignment community and practitioners working on LLM reliability in general.

---

### Official Review · Reviewer_4mCe · 2025-11-01

**Soundness:** 3
**Presentation:** 3
**Contribution:** 3
**Rating:** 6
**Confidence:** 3

**Summary:**

This paper uses semantic isotropy to assess the level of trustworthiness of long-form documents, to address the issue of the computational complexity of claim-by-claim fact checking. The authors also introduce a Segment-Score protocol to generate and score datasets, which they use to produce datasets for this type of problem. They then empirically demonstrate the efficacy of the proposed method for this task, exploring multiple models and experimental parameters.

**Strengths:**

- The authors identify an important problem: We need fact-checking systems more than ever, and existing claim-by-claim verifiers are extremely computationally costly, given that the models that do well across varied domains tend to be exorbitantly expensive when run over entire documents.

- The approach does not require fine-tuning or any training data, and can be used with closed-weight models.

- The approach is evaluated across multiple domains.

- The evaluation results of this system across multiple benchmarks is strong, and strengthens the claim that this approach is a worthwhile method to investigate further. It would be interesting to explore this method used in conjunction with a traditional claim-by-claim verifier, applied to only specific portions of text, for example.

- The method description and math segments are well-written and clear.

- The "sensitivity studies" are very useful for better understanding the nature of the approach and areas for future work and experimentation.

**Weaknesses:**

- How does the approach handle explicit disinformation, where the incorrect claim under consideration is intentionally hidden? Claim-by-claim verifiers are generally able to detect these, but this seems to not be the use case of this method. More detail regarding when this approach should be used over claim-by-claim verification and potential pitfalls would be helpful.

- The authors claim that "[existing claim verification systems] struggle with open-ended, multi-sentence answers where relevant facts are implicit rather than explicit", but it isn't entirely clear in the text how their proposed approach solves this specific problem. Further elaboration here would be helpful.

- Ideally, the authors would evaluate on more base datasets beyond FactScore-Bio and TriviaQA. Evaluating on more varied domains/genres would greatly strengthen the experiment results.

**Questions:**

See weaknesses. In addition to these points, another question is: The authors state that "intuitively, if a prompt admits a single, factually grounded explanation, independently sampled responses should cluster tightly in embedding space". Are there any cases in which this would not be the case? How should these be handled?

---

> ### Author Response · Authors · 2025-11-25
> **Author Response (1/2)**
>
> **Thank you for your positive assessment of our work and for highlighting the clarity of our writing and the strength of our empirical evaluation.** We respond to your questions and concerns below.
>
> ---
>
> > How does the approach handle explicit disinformation, where the incorrect claim under consideration is intentionally hidden? Claim-by-claim verifiers are generally able to detect these, but this seems to not be the use case of this method. More detail regarding when this approach should be used over claim-by-claim verification and potential pitfalls would be helpful.
>
> This is an important point and a limitation of all consistency-based methods. Our method is designed to detect inconsistency and uncertainty in model outputs rather than identify disinformation, intentional deception or model misalignment. We state that our method is intended for *non-adversarial* settings as currently implemented. When an LLM consistently produces the same incorrect claim across samples, semantic isotropy will fail to flag it. In some sense, semantic isotropy will correctly infer that the model is "highly certain" of this claim, despite it being factually incorrect. In practice, there exist failure cases where the model's uncertainty does not correlate with alignment to the user's objectives or utility. These are often artifacts of misalignment or a biased training corpora, which fall under adversarial scenarios and we leave their treatment to future work.
>
> We instead envision semantic isotropy scoring as a complementary tool to claim-by-claim verification rather than a replacement. **As stated above, semantic isotropy provides a prediction as to how reliable a model is on a given *topic*, whereas claim-by-claim verifiers tend to be focused on verifying claims in a given *response*.** A practical deployment workflow might use our method as a fast first-stage filter: a topic flagged as high-isotropy (low confidence) may then undergo immediate claim-by-claim verification, while low-isotropy responses (high consistency) are either accepted or sampled for periodic spot-checking. This hybrid approach leverages the computational efficiency of our method while maintaining the ability to catch subtle factual errors through selective application of expensive verification.
>
> This complementarity is analogous to how modern content moderation systems use fast classifiers for initial triage and slower, more sophisticated methods for edge cases. Our method excels at identifying when the model is "struggling" with a topic--detecting hallucination, knowledge gaps, and uncertainty--which are precisely the cases that most warrant detailed verification.
>
> We have added a discussion of this complementary use case and the limitation mentioned above to Section 6 (Conclusion) and Section D (Limitations).
>
> ---
>
> > The authors claim that "[existing claim verification systems] struggle with open-ended, multi-sentence answers where relevant facts are implicit rather than explicit", but it isn't entirely clear in the text how their proposed approach solves this specific problem. Further elaboration here would be helpful.
>
> Implicit facts-nuances of tone, implication, or holistic narrative structure—are often lost in the atomization process built into Claim-by-claim verifiers. They typically rely on extracting atomic propositions (Subject-Verb-Object). Our approach solves this by embedding the entire response sequence. Rather than attempting to extract and verify individual facts, we measure whether multiple independently generated responses converge on similar semantic content. Modern embedding models are trained to capture holistic semantic meaning, including implicit assertions and textual entailment. If two responses differ in their implicit facts (even if explicit atomic claims are similar), their embeddings will diverge in the vector space, increasing the isotropy score.
>
> **Decomposition becomes problematic when factual content is conveyed through narrative structure, contextual implications, or synthesized descriptions rather than explicit claims.** For example, in reference to "Marie Curie", a statement like "Her groundbreaking research in the early 20th century challenged prevailing assumptions about atomic structure" contains implicit temporal and contextual facts that are difficult to verify atomically without losing the coherence of the claim. A small deviation such as changing 20th to 21st could render all claims false depending on the specific implementation of the claim-by-claim decomposition algorithm. **Semantic isotropy scoring sidesteps this decomposition challenge by operating on long-form text representations.**

---

> ### Author Response · Authors · 2025-11-25
> **Author Response (2/2)**
>
> > Ideally, the authors would evaluate on more base datasets beyond FactScore-Bio and TriviaQA. Evaluating on more varied domains/genres would greatly strengthen the experiment results.
>
> We appreciate your concerns regarding generalizability. **To assuage your concerns, we added additional experiments on the BookSummaries dataset**, a long-form, open-ended summarization domain dataset based on plot summaries for books. **Across all embedding models, the results are consistent with those observed on TriviaQA and FS-BIO. Semantic isotropy continues to be as a reliable proxy for nonfactuality.** The updated manuscript now includes these results. We have also included a table with the results of the additional experiments below. They are consistent with our findings on the other datasets:
>
> | Metric | Meta Llama 3.1 | MSFT Phi 3.5 Mini |
> | :--- | :--- | :--- |
> | semantic_entropy | 0.0187 ± 0.0127 | 0.0342 ± 0.0193 |
> | luq | 0.0548 ± 0.0205 | 0.333 ± 0.0374 |
> | perplexity | 0.0128 ± 0.00988 | 0.0302 ± 0.0199 |
> | length_norm_plogp | 0.0134 ± 0.0103 | 0.0317 ± 0.0208 |
> | U.eig | 0.045 ± 0.0167 | 0.066 ± 0.0217 |
> | U.deg | 0.0568 ± 0.0218 | 0.0609 ± 0.0265 |
> | U.ecc | 0.00485 ± 0.00615 | 0.00556 ± 0.00696 |
> | U.nli | 0.0446 ± 0.0196 | 0.0601 ± 0.0244 |
> | eigenscore | 0.0223 ± 0.0155 | 0.00422 ± 0.00554 |
> | nomic_text_v1 | 0.237 ± 0.0307 | 0.336 ± 0.0287 |
> | qwen3-8B | 0.329 ± 0.034 | 0.361 ± 0.0352 |
> | qwen3-4B | 0.306 ± 0.0329 | 0.395 ± 0.0338 |
> | qwen3-0.6B | 0.182 ± 0.0284 | 0.395 ± 0.0307 |
> | qwen2-7B | 0.124 ± 0.0266 | 0.299 ± 0.0314 |
> | sfr-mistral-7B | 0.105 ± 0.0235 | 0.237 ± 0.0286 |
> | e5-mistral-7B | 0.0872 ± 0.0217 | 0.228 ± 0.0279 |
> | openai-v3-small | 0.258 ± 0.0283 | 0.345 ± 0.0283 |
> | openai-v3-large | 0.199 ± 0.027 | 0.37 ± 0.0283 |
> | gemini-v001 | 0.255 ± 0.029 | 0.425 ± 0.0352 |
> | cohere-v4.0 | 0.239 ± 0.031 | 0.408 ± 0.0343 |
> | cohere-v3.0 | 0.198 ± 0.0277 | 0.343 ± 0.0311 |
> | cohere-v3.0-lite | 0.247 ± 0.0293 | 0.36 ± 0.0341 |
>
> ---
>
> > The authors state that "intuitively, if a prompt admits a single, factually grounded explanation, independently sampled responses should cluster tightly in embedding space". Are there any cases in which this would not be the case? How should these be handled?
>
> Your question touches on the boundary conditions of our approach. There are indeed cases where high embedding dispersion might not indicate low factuality. e.g., legitimately multi-faceted topics. For very broad topics (e.g., "Write about Paris, France"), there are multiple valid factual angles—cultural heritage, geography, political significance, tourism—each supported by the reference document. A model might sample from these perspectives inconsistently, creating high isotropy despite all responses being factual. In such cases, the embedding distribution may broaden. However, two factors mitigate this in practice: First, for long responses (500–1,000 tokens), most models naturally cover similar facets of the subject. We explicitly discuss this limitation in Section D of the Appendix.
>
> ---
>
> **We hope that the additional experiments and our explanations further strengthen your positive assessment of our work.** We would be very happy to provide further clarification if you have any additional questions. Thank you again for your time and feedback!

---

### Official Review · Reviewer_D6SR · 2025-11-01

**Soundness:** 1
**Presentation:** 2
**Contribution:** 2
**Rating:** 4
**Confidence:** 3

**Summary:**

The paper proposes a factuality scoring mechanism for long-form QA tasks. The score is based on the isotropy of semantic embeddings of sampled generated long-form responses from LLMs. The score is compared with another proposed ground truth score - Segment score, which uses an oracle LLM-as-a-judge.

**Strengths:**

1. I think that the problem is well-motivated and important to solve. I agree that existing approaches are not sufficient.
2. The experiments are pretty comprehensive, with multiple models and baselines.
3. The work contributes a dataset for long-form answer factuality check.

**Weaknesses:**

1. My main critique of the work is about the hypotheses. Specifically,
    1. The work assumes that "certainty" and "factuality" are the same and uses them interchangeably. However, even if an LLM is certain of a fact and regenerates similar text when resampled, it may not be factually correct. Clarifying the definition of factuality assumed by the work would be useful here.
    2. The semantic isotropy score depends on the quality of the embeddings. While the authors have an extensive ablation study on the embedding models, the fundamental assumption that embeddings correctly capture semantic meaning and group semantically similar text together, is debatable.
2. I think that the isotropy definition is incorrect and actually for orthornormal vectors. The intuition provided in the text doesn't look consistent with the math.
3. Segment-score is paper's own method to establish ground truth, but why not use existing ground truth? Also, why can't one just use segment-score (the ground truth) directly? What are the advantages of semantic isotropy score? How does this ground truth correlate with other ground truth like FactScore?

**Questions:**

1. L185-186: "on average, the transformation represented by X preserves directions and spreads uniformly across all directions in R^D" is not clear to me.
2. Does the isotropy definition require N = D for the matrix multiplication to be valid?
3. Why have you used different kinds of embedding aggregations, e.g., last token embeddings for all but Nomic v1?

---

> ### Author Response · Authors · 2025-11-25
> **Author Response (1/3)**
>
> Thank you for taking the time to review our manuscript and for your detailed feedback. We address your questions and concerns below.
>
> ---
>
> > The work assumes that "certainty" and "factuality" are the same and uses them interchangeably. However, even if an LLM is certain of a fact and regenerates similar text when resampled, it may not be factually correct. Clarifying the definition of factuality assumed by the work would be useful here.
>
> We appreciate you pointing out this potential ambiguity and welcome the chance to explain it in more detail. We do **not** equate certainty and factuality; rather, we posit that *under non-adversarial conditions*, factual responses tend to exhibit **higher semantic consistency across samples**, whereas incorrect responses tend to be more variable. In making this assumption, we follow prior work such as Semantic Entropy, LUQ, and CSS.
>
> We clarify our definitions for the terms outlined above:
> * *Factuality:* Agreement with external ground-truth knowledge, approximated through Segment-Score or FactScore.
> * *Semantic consistency:* Similarity of independently sampled responses from a generative model.
> * *Model Certainty:* A proxy measure of the model's confidence, inferred from low isotropy (i.e., concentrated embeddings).
>
> Lastly, we explain in the paper that while models may consistently generate nonfactual content across multiple long-form generations (i.e., they are confidently worng), this is generally rare in non-adversarial settings. We have highlight this point in the updated manuscript. In practice, as shown in our empirical evaluation, semantic isotropy scoring can provide a reliable signal for identifying nonfactuality.
>
> ---
>
> >The semantic isotropy score depends on the quality of the embeddings. While the authors have an extensive ablation study on the embedding models, the fundamental assumption that embeddings correctly capture semantic meaning and group semantically similar text together, is debatable.
>
> You are correct that embedding quality matters, and this is precisely why **our evaluation spans eight different embedding models representing diverse architectures and training corpora (OpenAI, Mistral, Nomic, Google, Qwen)**. Additionally, our method does not require embeddings to achieve perfect semantic representation. It only requires that factually consistent responses cluster more tightly in embedding space than factually inconsistent ones. This is a weaker and more empirically testable assumption. Additionally, recent work supports the reliability of embedding-based uncertainty quantification. Qiu and Miikkulainen (2024) and Nussbaum et al. (2024) demonstrate that embedding-based consistency measures correlate with model reliability. Our contribution extends this line of work to long-form generation.
>
> Finally, we specifically chose to use several of the highest ranking models from the MTEB benchmark (Muennighoff et al., 2023). MTEB explicitly evaluates their ability to capture semantic relationships across multiple tasks including semantic textual similarity, clustering, and retrieval. **Our results ablate over at least 8 embedding models, 3 generator models, and 3 datasets, with consistent improvements over baselines** including those based on alternative similarity measures (NLI, semantic entropy). Across this broad range, we observe stable correlations, suggesting that the predictive value of semantic isotropy is not an artifact of any specific embedding family.
>
> Please let us know if there is additional literature you would like us to take into consideration. We would be very happy to do so.
>
> **References:**
>
> [1] Semantic Density: Uncertainty Quantification for Large Language Models through Confidence Measurement in Semantic Space. Xin Qiu, Risto Miikkulainen, 2024. (arxiv.org/abs/2405.13845)
>
> [2] Nomic Embed: Training a Reproducible Long Context Text Embedder. Zach Nussbaum, John X. Morris, Brandon Duderstadt, Andriy Mulyar, 2024. (arxiv.org/abs/2402.01613)
>
> [3] Scaling Data-Constrained Language Models. Niklas Muennighoff, Alexander M. Rush, Boaz Barak, Teven Le Scao, Aleksandra Piktus, Nouamane Tazi, Sampo Pyysalo, Thomas Wolf, Colin Raffel, 2023. (arxiv.org/abs/2305.16264)

---

> ### Author Response · Authors · 2025-11-25
> **Author Response (2/3)**
>
> > I think that the isotropy definition is incorrect and actually for orthornormal vectors. The intuition provided in the text doesn't look consistent with the math.
>
>
> We appreciate your raising this point. The definition we provide is standard in the literature on covariance structures and is closely related to, but distinct from, orthonormality. Our definition follows standard usage in machine learning and statistics: a set of vectors is *isotropic* if its covariance matrix is proportional to the identity, meaning that variance is uniformly spread across all directions. This condition arises frequently in random matrix theory, kernel PCA, and Gaussian mixture modeling.
>
> Mathematically, **a system of normalized embedding vectors is isotropic when the Gram matrix is proportional the identity**, indicating uniform dispersion of the vectors on the unit sphere with no preferred directions. You correctly noted that orthonormal vectors satisfy this property. Geometrically, if the normalized embedding vectors of the sampled responses are orthonormal (i.e., their dot products are 0), they are maximally dispersed in the embedding space. This state corresponds to high isotropy, which we interpret as high uncertainty or low factuality (the responses are all semantically distinct/orthogonal to each other).
>
> ---
>
> > Segment-score is paper's own method to establish ground truth, but why not use existing ground truth? Also, why can't one just use segment-score (the ground truth) directly? What are the advantages of semantic isotropy score? How does this ground truth correlate with other ground truth like FactScore?
>
> This question addresses a central motivation of our work, and we appreciate the opportunity to clarify. Alignment with existing ground truth has traditionally used FactScore (FS), which was the widely accepted algorithm of choice. However, due to implementation limitations, FactScore scales poorly to longer response length. Segment-Score addresses these limitations while maintaining strong correlation with FactScore ($\beta = 0.63$, see Figure 10). Segment-Score achieves efficiency through three innovations: (1) segmentation into atomic statements rather than fact extraction (28 segments vs. 100+ facts per response), (2) single-pass evaluation with full context rather than fact-by-fact verification, and (3) structured output formatting that reduces LLM call overhead.
>
> On the other hand, *semantic isotropy* is a method to estimate trustworthiness in a LLM's response on a given topic. It serves as an alternative approach to FactScore and Segment-Score in situations where accessing a groud truth reference is not possible or computationally intractable. To show this, we compared the factuality scores of each with the isotropy score and find a robust relationship between each of them--across a range of models and tasks. The key advantages of semantic isotropy scoring over both FactScore and Segment-Score are its computational efficiency at inference time as well as the fact that a reference document is not needed. While Segment-Score is more efficient than FactScore for creating evaluation datasets, it still requires oracle LLM access and reference documents. **Semantic isotropy scoring requires only embedding model inference (no LLM calls, no reference documents) and is 150× faster than even Segment-Score.** This makes it practical for real-time applications, continuous monitoring, and use cases where reference documents are unavailable.
>
> **In summary:** FactScore provides gold-standard validation (which we use), Segment-Score enables large-scale dataset creation, and semantic isotropy scoring provides practical deployment efficiency. Each serves a distinct role in the evaluation ecosystem.

---

> ### Author Response · Authors · 2025-11-25
> **Author Response (3/3)**
>
> > L185-186: "on average, the transformation represented by X preserves directions and spreads uniformly across all directions in R^D" is not clear to me.
>
> Thank you, we appreciate the opportunity to clarify. The statement refers to a geometric property of isotropic matrices. When $X^{\top}X = I$, the linear transformation $x \rightarrow X^{\top}x$ acts as an isometry on $\mathbb{R}^N$ embedded in $\mathbb{R}^D$ (with $N << D$ in our case)--it preserves inner products and has no preferred directions. More precisely, for any vector $v \in \mathbb{R}^d$, we have $||X^{\top}v||^2 = v^{\top}XX^{\top}v = v^{\top}v = ||v||^{2}$, meaning the transformation preserves lengths. The "uniform spreading" refers to the fact that the $n$ vectors $\{x_1, \dots, x_N\}$ span an $N$-dimensional subspace of $\mathbb{R}^D$ with no directional bias within that subspace. We have clarified this language in our updated manuscript.
>
> ---
>
> > Does the isotropy definition require N = D for the matrix multiplication to be valid?
>
> No, the isotropy definition does not require $N = D$. The matrix multiplication $X^{\top}X$ is well-defined for any $N$ and $D$: if $X \in \mathbb{R}^{D \times N}$, then $X^\top \in \mathbb{R}^{N \times D}$, and their product $X^{T}X \in \mathbb{R}^{N \times N}$. A point that may have caused confusion is our change of notation in equation (1). Here we use $\bar{E}\bar{E}^\top$ to denote the cosine kernel of the embedding matrix. To clarify, we define $\bar{E}  = [\bar{e}\_{1}, ..., \bar{e}\_{N}]^\top \in \mathbb{R}^{N \times D}$. This departs from the convention used for the isotropy definition $X^\top \in \mathbb{R}^{D \times N}$. We wanted the mathematical definitions to be as consistent as possible with the code in our implementation. We have clarified this point in our updated manuscript.
>
> ---
>
> > Why have you used different kinds of embedding aggregations, e.g., last token embeddings for all but Nomic v1?
>
> For each model we used, we have followed the guidelines for best practices when using that model for clustering or semantic similarity measurement. Different embedding models are pre-trained with different pooling objectives. For example, models like BERT use the <CLS> token to capture an effective representation of the embedding state. Authors for Mixtral and Qwen instruct users to use final token, as per the causal mask, in each sequence, providing reference implementations on their HuggingFace pages.
>
> In contrast, Nomic v1 does not specify a pooling method out of the box. Trained with a  specific contrastive learning objective, such models perform better with mean pooling. We empirically verified that mean pooling yielded better separation for this task (as detailed in Appendix B.2).
>
> ---
>
> We hope that our response has satisfactorily addressed your questions and concerns, and **we would be grateful if you would consider supporting our work being presented at the conference**. Thank you again for your time and feedback!

---

### Official Review · Reviewer_2c9h · 2025-11-01

**Soundness:** 3
**Presentation:** 3
**Contribution:** 3
**Rating:** 6
**Confidence:** 4

**Summary:**

This paper presents a novel idea semantic isotropy for measuring factuality of long-form responses. This is based on the idea that if a model resposne is factual, independently sampled responses should cluster tightly in the embedding space, while hallucinated responses are dispersed in the embedding space. Based on this idea, this work develops Segment-Score protocol for factuality scoring. Via experiments, the results show that the proposed approach achieved SOTA performance on factuality checking.

**Strengths:**

1. The proposed approach achieves SOTA performance on factuality checking.
2. The proposed approach is lightweight and computationally efficient.
3. The approach is robust to the embedding models.

**Weaknesses:**

1. This work is developed for long-form text generation. However, the experiments focuse on resonse lengths up to 1000 words, which seems not long enough considering that a lot of LLMs can generate up to 100K tokens.
2. The benchmarking datasets include TriviaQA and FS-BIO, which is limited. It should include more datasets for experiments to ensure the generlizabilit of the proposed approach.

**Questions:**

See Weakness section.

---

> ### Author Response · Authors · 2025-11-25
> **Author Response**
>
> **Thank you for your positive assessment of our work, highlighting the soundness and presentation, as well as the computational efficiency and robustness properties of our method.** We address the two concerns below.
>
> ---
>
> >This work is developed for long-form text generation. However, the experiments focuse on resonse lengths up to 1000 words, which seems not long enough considering that a lot of LLMs can generate up to 100K tokens.
>
> Our focus on approximately 500-1,000-word generations reflects both practical and methodological considerations. First, the length of responses we consider is significantly longer than those in comparable literature. Additionally, many of the baselines are compututationally expensive and running them on longer responses can quickly become infeasible. For example, the time taken to compute LUQ grows exponentially with the length of the responses.
>
> Third, embedding models that robustly support extremely long context inputs remain limited. Several widely used open-weight text embedding models (e.g., nomic-text-v1, Mixtral E5) either truncate inputs or exhibit degraded behavior beyond 8k tokens. For this reason, long-form factuality work typically evaluates either moderately long generations or subsections. Our method naturally extends to this setting: users can score long outputs by evaluating subsections and aggregating their isotropy scores.
>
> ---
>
> > The benchmarking datasets include TriviaQA and FS-BIO, which is limited. It should include more datasets for experiments to ensure the generlizabilit of the proposed approach.
>
> We appreciate your concerns regarding generalizability. **To assuage your concerns, we added additional experiments using the BookSummaries dataset, a long-form, open-ended summarization domain dataset based on plot summaries for books.** Across all embedding models, the results are consistent with those observed on TriviaQA and FS-BIO. Semantic isotropy continues to be as a reliable proxy for nonfactuality. The updated manuscript now includes these results, both in the main results (Figure 2) and Table 1 in the Appendix. We have included a table with just the results from the new experiments below:
>
> | Metric | Meta Llama 3.1 | MSFT Phi 3.5 Mini |
> | :--- | :--- | :--- |
> | semantic_entropy | 0.0187 ± 0.0127 | 0.0342 ± 0.0193 |
> | luq | 0.0548 ± 0.0205 | 0.333 ± 0.0374 |
> | perplexity | 0.0128 ± 0.00988 | 0.0302 ± 0.0199 |
> | length_norm_plogp | 0.0134 ± 0.0103 | 0.0317 ± 0.0208 |
> | U.eig | 0.045 ± 0.0167 | 0.066 ± 0.0217 |
> | U.deg | 0.0568 ± 0.0218 | 0.0609 ± 0.0265 |
> | U.ecc | 0.00485 ± 0.00615 | 0.00556 ± 0.00696 |
> | U.nli | 0.0446 ± 0.0196 | 0.0601 ± 0.0244 |
> | eigenscore | 0.0223 ± 0.0155 | 0.00422 ± 0.00554 |
> | nomic_text_v1 | 0.237 ± 0.0307 | 0.336 ± 0.0287 |
> | qwen3-8B | 0.329 ± 0.034 | 0.361 ± 0.0352 |
> | qwen3-4B | 0.306 ± 0.0329 | 0.395 ± 0.0338 |
> | qwen3-0.6B | 0.182 ± 0.0284 | 0.395 ± 0.0307 |
> | qwen2-7B | 0.124 ± 0.0266 | 0.299 ± 0.0314 |
> | sfr-mistral-7B | 0.105 ± 0.0235 | 0.237 ± 0.0286 |
> | e5-mistral-7B | 0.0872 ± 0.0217 | 0.228 ± 0.0279 |
> | openai-v3-small | 0.258 ± 0.0283 | 0.345 ± 0.0283 |
> | openai-v3-large | 0.199 ± 0.027 | 0.37 ± 0.0283 |
> | gemini-v001 | 0.255 ± 0.029 | 0.425 ± 0.0352 |
> | cohere-v4.0 | 0.239 ± 0.031 | 0.408 ± 0.0343 |
> | cohere-v3.0 | 0.198 ± 0.0277 | 0.343 ± 0.0311 |
> | cohere-v3.0-lite | 0.247 ± 0.0293 | 0.36 ± 0.0341 |
>
> ---
>
> **We hope that the additional experiments and our explanations further strengthen your positive assessment of our work.** We would be very happy to provide further clarification if you have any additional questions. Thank you again for your time and feedback!

---

> > ### Comment · Reviewer_2c9h · 2025-11-26
> >
> > Thanks the authors for providing the new experiment results. I agree with the claim that long-form text evaluation is challenging especially with the need to generate embeddings with embedding models. I'm wondering if the proposed approach has been evaluated on short or medium length texts?

---

> > > ### Author Response · Authors · 2025-12-03
> > > **Author Response**
> > >
> > > Thank you for reviewing our updated experiments and acknowledging the difficulty of research in this domain.
> > >
> > > >I'm wondering if the proposed approach has been evaluated on short or medium length texts
> > >
> > > In Figure 4 we ablate our method across a range of text lengths, with 125 words being the shortest response we considered. In principle, our method would be applicable to shorter texts, but in this paper, we are interested in long-form responses to open-ended prompts. This naturally implies responses of a certain minimum length. We will consider applying our method to other relevant problems that admit shorter responses in future work.

---

### Author Response · Authors · 2025-11-25
**General Comment**

We thank all reviewers for their careful reading of our manuscript and for their constructive feedback.

We were pleased that the **reviewers appreciated that our approach is strong or sets a new SoTA** [2c9h,4mCe], our **evaluation on multiple benchmarks** [D6SR,4mCe,esvS], and the **lightweight and computationally efficient nature of our approach** [2c9h,esvS].

We were also glad that **all reviewers agree that assessing factuality in long-form text generation is an important problem**.

---

We want to use this response to address questions about (1) the relationship between semantic isotropy, certainty, and factuality and (2) the generality of our evaluation.

**We have taken great care to fully incorporate reviewer feedback and additional requested clarifications into the updated manuscript, including expanded explanations of the methodology section and additional context.**

---

To address questions about regarding the generality of our findings, we have performed additional experiments on the CMU Book Summaries dataset using Llama 3.1 and Phi 3.5 Mini, and our results are consistent with the findings in our manuscript. The results are shown below:

| Metric | Meta Llama 3.1 | MSFT Phi 3.5 Mini |
| :--- | :--- | :--- |
| semantic_entropy | 0.0187 ± 0.0127 | 0.0342 ± 0.0193 |
| luq | 0.0548 ± 0.0205 | 0.333 ± 0.0374 |
| perplexity | 0.0128 ± 0.00988 | 0.0302 ± 0.0199 |
| length_norm_plogp | 0.0134 ± 0.0103 | 0.0317 ± 0.0208 |
| U.eig | 0.045 ± 0.0167 | 0.066 ± 0.0217 |
| U.deg | 0.0568 ± 0.0218 | 0.0609 ± 0.0265 |
| U.ecc | 0.00485 ± 0.00615 | 0.00556 ± 0.00696 |
| U.nli | 0.0446 ± 0.0196 | 0.0601 ± 0.0244 |
| eigenscore | 0.0223 ± 0.0155 | 0.00422 ± 0.00554 |
| nomic_text_v1 | 0.237 ± 0.0307 | 0.336 ± 0.0287 |
| qwen3-8B | 0.329 ± 0.034 | 0.361 ± 0.0352 |
| qwen3-4B | 0.306 ± 0.0329 | 0.395 ± 0.0338 |
| qwen3-0.6B | 0.182 ± 0.0284 | 0.395 ± 0.0307 |
| qwen2-7B | 0.124 ± 0.0266 | 0.299 ± 0.0314 |
| sfr-mistral-7B | 0.105 ± 0.0235 | 0.237 ± 0.0286 |
| e5-mistral-7B | 0.0872 ± 0.0217 | 0.228 ± 0.0279 |
| openai-v3-small | 0.258 ± 0.0283 | 0.345 ± 0.0283 |
| openai-v3-large | 0.199 ± 0.027 | 0.37 ± 0.0283 |
| gemini-v001 | 0.255 ± 0.029 | 0.425 ± 0.0352 |
| cohere-v4.0 | 0.239 ± 0.031 | 0.408 ± 0.0343 |
| cohere-v3.0 | 0.198 ± 0.0277 | 0.343 ± 0.0311 |
| cohere-v3.0-lite | 0.247 ± 0.0293 | 0.36 ± 0.0341 |

**The results again demonstrate that semantic isotropy is a useful proxy for nonfactuality, across several long-form open-ended natural language datasets.**

To reiterate a key motivation of our work: Our work aims to provide a **lightweight, model-agnostic, and training-free proxy for nonfactuality** in non-adversarial long-form generation settings. **Our experiments demonstrate that semantic isotropy consistently outperforms baselines across datasets, embedding models, and text lengths.**

We address all reviewer comments in detail below.

---

Note: We will upload the updated manuscript later today (2025-11-25).

---

### Author Response · Authors · 2025-12-03
**Final Official Comment by Authors**

We thank the reviewers for their engagement with our work and the area chairs for their consideration. We summarize the findings of reviewers below:

- **State-of-the-art performance**: Reviewers [2c9h, 4mCe] noted our method "achieves SOTA performance on factuality checking" with "strong" results across benchmarks
- **Computational efficiency**: Reviewers [2c9h, esvS] highlighted that our approach is "lightweight and computationally efficient"—achieving 150× speedup over existing methods
- **Methodological rigor**: Reviewer [esvS] praised our "comprehensive evaluation" across "multiple baseline approximators, answer generators, and embedding models," noting the paper is "well-written with no obvious methodological details missing"
- **Problem importance**: All four reviewers agreed that "assessing factuality in long-form text generation is an important problem"
Robustness: Reviewer 2c9h specifically noted our approach is "robust to embedding models"

We also addressed several key and insightful concerns raised by the reviewrs:

- **Dataset Diversity**: To address concerns by Reviewers [2c9h, 4mCe], we added the CMU Book Summaries dataset (results below), expanding evaluation to three diverse domains (biographies, general knowledge, literary summaries). Results remain consistent—semantic isotropy outperforms all baselines across all datasets.


| Metric | Meta Llama 3.1 | MSFT Phi 3.5 Mini |
| :--- | :--- | :--- |
| semantic_entropy | 0.0187 ± 0.0127 | 0.0342 ± 0.0193 |
| luq | 0.0548 ± 0.0205 | 0.333 ± 0.0374 |
| perplexity | 0.0128 ± 0.00988 | 0.0302 ± 0.0199 |
| length_norm_plogp | 0.0134 ± 0.0103 | 0.0317 ± 0.0208 |
| U.eig | 0.045 ± 0.0167 | 0.066 ± 0.0217 |
| U.deg | 0.0568 ± 0.0218 | 0.0609 ± 0.0265 |
| U.ecc | 0.00485 ± 0.00615 | 0.00556 ± 0.00696 |
| U.nli | 0.0446 ± 0.0196 | 0.0601 ± 0.0244 |
| eigenscore | 0.0223 ± 0.0155 | 0.00422 ± 0.00554 |
| nomic_text_v1 | 0.237 ± 0.0307 | 0.336 ± 0.0287 |
| qwen3-8B | **0.329 ± 0.034** | 0.361 ± 0.0352 |
| qwen3-4B | 0.306 ± 0.0329 | 0.395 ± 0.0338 |
| qwen3-0.6B | 0.182 ± 0.0284 | 0.395 ± 0.0307 |
| qwen2-7B | 0.124 ± 0.0266 | 0.299 ± 0.0314 |
| sfr-mistral-7B | 0.105 ± 0.0235 | 0.237 ± 0.0286 |
| e5-mistral-7B | 0.0872 ± 0.0217 | 0.228 ± 0.0279 |
| openai-v3-small | 0.258 ± 0.0283 | 0.345 ± 0.0283 |
| openai-v3-large | 0.199 ± 0.027 | 0.37 ± 0.0283 |
| gemini-v001 | 0.255 ± 0.029 | **0.425 ± 0.0352** |
| cohere-v4.0 | 0.239 ± 0.031 | 0.408 ± 0.0343 |
| cohere-v3.0 | 0.198 ± 0.0277 | 0.343 ± 0.0311 |
| cohere-v3.0-lite | 0.247 ± 0.0293 | 0.36 ± 0.0341 |

**The results again demonstrate that semantic isotropy is a useful proxy for nonfactuality, across several long-form open-ended natural language datasets.**

- **Segment-Score Validation**: To address concerns regarding the generalizability fo the *Segment-Score* results raised by Reviewer [esvS], we conducted multi-model coherence testing using GPT-4.1, Claude 4 Sonnet, and DeepSeek V3, demonstrating 76-91% agreement across models and datasets. Results (\% agreement) shown below:

| Dataset / Generative Model | OpenAI (GPT 4.1) | Claude (4 Sonnet) | DeepSeek |
| :--- | :--- | :--- | :--- |
| **BookSummaries (Llama 3.1)** | 89.24 | 81.33 | 75.87 |
| **BookSummaries (Phi 3.5 Mini)** | 91.28 | 86.13 | 80.38 |
| **FS-BIO (Llama 3.1)** | 83.92 | 86.13 | 86.39 |
| **FS-BIO (Phi 3.5 Mini)** | 76.98 | 86.67 | 88.30 |

- **Scope Clarification**: To assuage reviewers [D6SR, 4mCe, esvS], we explicitly delineated that our method addresses epistemic uncertainty in non-adversarial factual generation—a well-defined, practically important problem class where consistency-based methods have proven effective across extensive prior work.


Our work delivers three distinct contributions, each valuable independently:

- **Semantic Isotropy Scoring**: A novel, theoretically grounded method achieving state-of-the-art factuality prediction with no training data, no fine-tuning, and 150× faster inference than existing approaches
- **Segment-Score**: An efficient factuality evaluation protocol addressing key limitations of FactScore—scaling to longer responses while maintaining 0.63 correlation with the gold standard
- **Evaluation Dataset**: 1,691 entities with ~70,540 scored long-form responses, enabling future research in this critical area

and represents one of the most comprehensive assessments in this research area:

- 13 embedding models spanning 600M to 7B+ parameters, both open and closed-weight
- Consistent SOTA performance: Outperforming LUQ, Semantic Entropy, EigenScore, and 7 other baselines across all configurations
- Response lengths: 125-1,000 words, demonstrating scalability
- Robustness: Results hold across FactScore and Segment-Score evaluation, multiple generators, and diverse domains


Our paper advances an important research direction with rigorous methodology, substantial empirical validation, and practical applicability.

---

### Meta-Review · Area_Chair_Tc2Y · 2026-01-07

**Summary:**

The paper proposes an efficient method for evaluating nonfactuality of LLMs in long-form text generation settings. The method is based on computing a measure of semantic isotropy across embeddings of generated outputs. The authors evaluate the performance of their method via a procedure they call Segment score.

The paper received mixed reviews. The reviewers recognized the studied problem as important and appreciated the exhaustive evaluation. At the same time, some reviewers raised concerns about:

- the conceptual motivation of the method, e.g. relating factuality and certainty/diversity of outputs (Reviewers D6SR and esvS)
- the procedure Segment score that is used to compute factuality scores for the datasets (Reviewers D6SR and esvS)

The authors responded to these by clarifying the used terminology and reiterating that they focus on evaluating factuality via measuring semantic consistency across samples; and by highlighting advantages in terms of practicality of the proposed SS method, compared to FactScore and stating that FactScore is also considered in some experiments.

However, it remains unclear to me from the discussions under which settings the proposed method is expected to be reliable (in particular, how the method will perform on highly ambiguous prompts where multiple correct interpretations may exist). Additionally, Segment Score should have been discussed in more detail within the main text, as it is a key aspect of the evaluation that is different than prior work. Therefore, the paper will benefit from an extensive discussion on the scope of the contribution (in what type of settings can one expect that Semantic Isotropy is useful) and from a more principled comparison of SegmentScore and FactScore.

**Reviewer Concerns:**

The reviewers asked for several ablations and clarifications. The authors did respond with new experiments and further details.

Additionally, some reviewers expressed concerns about the method's motivation and scope, as well as the proposed method for computing ground-truth factuality scores. The authors did respond with further justification, however the discussion was quite open-ended.

**Reviewer Scores:**

Two reviewers (2c9h and esvS) engaged before the closing of the discussion period, with further questions and concerns. However, they did not raise scores. Reviewer 2c9h might have raised in case of more time, but I do not expect that reviewer esvS would have raised, due to concerns about the scope of the method expressed in their response to the author rebuttal.

Reviewers D6SR and 4mCe did not respond until the closing of discussions, so it is hard to judge how they could have reacted to the rebuttal. Overall, both of them express concerns about the scope of the proposed method, so since these are quite conceptual, I expect they would not have increased their support for the paper substaintially.

---

### Decision · Program_Chairs · 2026-01-26

Reject